# INSEva: A Comprehensive Chinese Benchmark for Large Language Models in Insurance

## Abstract

Insurance, as a critical component of the global financial system, demands high standards of accuracy and reliability in AI applications. While existing benchmarks evaluate AI capabilities across various domains, they often fail to capture the unique characteristics and requirements of the insurance domain. To address this gap, we present INSEva, a comprehensive Chinese benchmark specifically designed for evaluating AI systems' knowledge and capabilities in insurance. INSEva features a multi-dimensional evaluation taxonomy covering business areas, task formats, difficulty levels, and cognitive-knowledge dimension, comprising 38,704 high-quality evaluation examples sourced from authoritative materials. Our benchmark implements tailored evaluation methods for assessing both faithfulness and completeness in open-ended responses. Through extensive evaluation of 9 state-of-the-art Large Language Models (LLMs), we identify significant performance variations across different dimensions. While general LLMs demonstrate basic insurance domain competency with average scores above 80, substantial gaps remain in handling complex, real-world insurance scenarios. The benchmark will be public soon.

## 1 Introduction

Insurance is one of the fundamental pillars of the global financial ecosystem (Pfeifer & Langen, 2021), performing irreplaceable infrastructure functions in economic stability worldwide. Recently, Artificial Intelligence (AI) has emerged as a transformative force in this field (Jaiswal, 2023). Due to the high risk of insurance decisions, AI systems are required not only to possess domain knowledge but also to ensure high accuracy and harmlessness. Therefore, specialized benchmarks for evaluating the domain-specific knowledge and capabilities of AI in insurance have become particularly crucial.

Many recent studies have attempted to construct benchmarks in the financial domain, which is the parent domain of insurance, yet few focus specifically on insurance. Existing financial benchmarks (Chen et al., 2021; Zhu et al., 2021; Chen et al., 2022; Shah et al., 2022; Zhu et al., 2024; Zhang et al., 2023; Peng et al., 2025; Xie et al., 2024) evaluate LLMs' capabilities within the broad financial domain. The financial domain centers on capital circulation, value appreciation, and financial instrument transactions, emphasizing wealth growth through capital operations. Insurance, however, focuses on risk management, risk assessment, pricing, and claims services under complex constraints (insurance clauses) to ensure policyholder protection and economic compensation during risk events, as shown in Figure 1. These fundamental differences require a dedicated insurance benchmark, while currently only a few of the benchmarks are proposed for insurance. InsQABench (Ding et al., 2025) collects insurance commonsense question-answering data for evaluation. INS-MMBench (Lin et al., 2024) focuses on evaluating cross-modal alignment capabilities in auto insurance scenarios. These benchmarks only include basic knowledge questions in insurance, which fail to adequately cover the insurance's unique risk characteristics, complex product clauses, deep causal reasoning, cross-domain knowledge integration (e.g., medical, legal), and critical elements in insurance sales processes.

To address these limitations, we present INSEva, a comprehensive Chinese benchmark specifically designed for evaluating AI systems' knowledge and capabilities in the insurance domain. Based on an in-depth study of the insurance production environment, we develop a multi-dimensional evaluation taxonomy that encompasses business areas, task formats, difficulty levels, and cognitive-

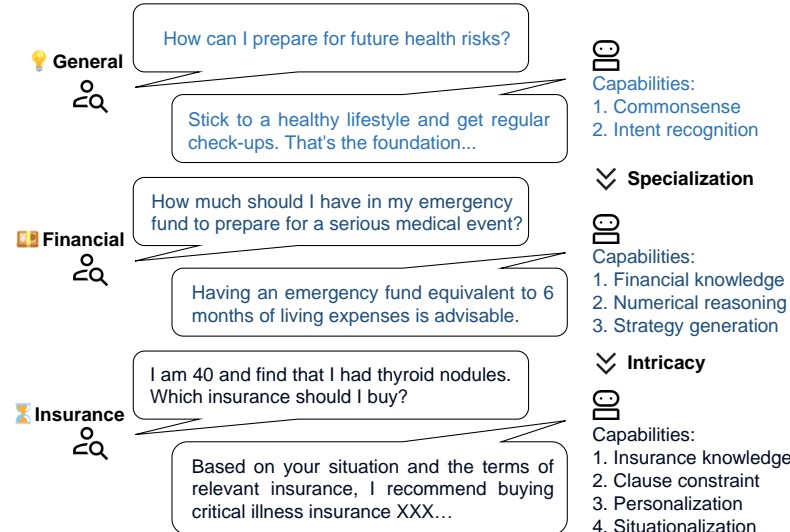

Figure 1: A comparison of question-answering examples and required model capabilities in the general, financial, and insurance domains. It shows how questions are becoming increasingly specialized and intricate in the general, financial, and insurance domains, and how this places different demands on LLM capabilities.

knowledge domains, thereby ensuring comprehensive assessment coverage. Our data construction pipeline incorporates three key stages: (1) collection from authoritative sources, (2) systematic data augmentation to enhance diversity while maintaining domain authenticity, and (3) rigorous quality control methods involving expert validation. This methodical approach yields 38,704 high-quality evaluation examples spanning the insurance domain. Given the high risk of insurance operations where errors can have significant financial and regulatory consequences, we implement tailored evaluation methods for different question types. For structured questions with definitive answers, we employ conventional deterministic metrics (primarily accuracy), while for open-ended responses, we deploy a specially designed LLM-based evaluation framework that systematically assesses both faithfulness to retrieved information and completeness to ground truth. Through these methodological innovations, INSEva not only provides a comprehensive assessment of insurance-specific capabilities but also effectively differentiates model performance across multiple dimensions. The benchmark thus offers valuable guidance for domain-specific development efforts and targeted model improvements in the domain.

Using INSEva, we evaluate 9 state-of-the-art LLMs (including closed-source, open-source, and domain-specific LLMs), revealing differentiated performance across various taxonomy dimensions and identifying numerous meaningful insights. While general LLMs demonstrate a foundational understanding of insurance concepts, as evidenced by average scores exceeding 80, a considerable gap remains between their capabilities and the expertise required to tackle complex, real-world insurance problems. Specifically, we observe a performance bottleneck in tasks requiring logical reasoning and a trade-off between faithfulness and completeness in generated content. Notably, the underperformance of the financial LLMs compared to general LLMs underscores the need for specialized benchmarks tailored to the unique demands of insurance, as existing financial benchmarks may not adequately capture the nuances of this critical domain. Furthermore, we fine-tune an LLM using data from our pipeline, achieving substantial performance gains, thereby validating the effectiveness of both our data construction methodology and the benchmark itself.

To summarize, our contributions are as follows:

- We propose a comprehensive Chinese benchmark for the insurance domain, with a multi-dimensional evaluation taxonomy and 38,704 high-quality evaluation examples from authoritative sources.

- We design a data construction and evaluation pipeline to ensure the high-quality of data and evaluation for the insurance domain.

- We evaluate leading LLMs to show their domain-specific gaps, and further validate our benchmark's utility by confirming it accurately reflects performance improvements from targeted fine-tuning.

## 2 RELATED WORKS

### 2.1 GENERAL BENCHMARKS

Many general benchmarks have been developed to assess the broad capabilities of models across diverse tasks and domains. For example, the GLUE benchmark (Wang et al., 2019) comprises a collection of natural language understanding tasks. MMLU (Hendrycks et al., 2021) evaluates multi-task language understanding across a wide range of subjects, assessing both knowledge acquisition and reasoning abilities. The HELM benchmark (Liang et al., 2023) adopts a holistic approach to evaluate models across a diverse set of scenarios and metrics, emphasizing real-world relevance and ethical considerations. BIG-bench (Srivastava A, 2023) introduces tasks that require reasoning, common sense, and cultural knowledge. While these general benchmarks provide valuable insights into the overall performance of LLMs, they often lack the domain-specific focus and granularity required to assess performance in specialized areas such as insurance, highlighting the need for tailored evaluation resources.

### 2.2 FINANCE & INSURANCE BENCHMARKS

As the parent domain of insurance, the development of financial benchmarks for financial models has accelerated in recent years, providing structured frameworks for assessing models' capabilities in the finance domain. FinQA (Chen et al., 2021) advances numerical reasoning evaluation by constructing a dataset requiring multi-step calculations and inference from financial reports. FLUE (Shah et al., 2022) introduces comprehensive financial language understanding tasks. CFLUE (Zhu et al., 2024) evaluates models through two main dimensions in the Chinese financial domain, including knowledge assessment and application assessment. FinBen (Xie et al., 2024) is an extensive open-source evaluation benchmark, including 36 datasets spanning 24 financial tasks. Despite the proliferation of financial evaluation resources, they are difficult to apply directly to insurance, which presents unique terminology, regulatory frameworks, and reasoning requirements distinct from those in general finance contexts. Some financial benchmarks (such as CFLUE) include subtasks in insurance, but the questions in these subtasks generally focus only on basic insurance knowledge, lacking real-world applications.

Compared with the financial domain, there are far fewer benchmarks specifically for insurance. InsuranceQA (Feng et al., 2015) is the first benchmark for insurance that contains data collected from the internet. InsQABench (Ding et al., 2025) encompasses three specialized insurance QA tasks, including insurance commonsense knowledge, insurance structured database, and insurance unstructured document. Ins-MMBench (Lin et al., 2024) comprises a total of 2.2k multimodal multiple-choice questions for evaluating large vision-language models. Due to the lack of a complete insurance taxonomy, these benchmarks fail to fully cover insurance tasks.

## 3 INSEVA BENCHMARK

### 3.1 OVERVIEW

INSEva is a comprehensive resource designed to evaluate the performance of models in the insurance-specific tasks. The benchmark encompasses a diverse set of samples and tasks, drawing data from authoritative Chinese sources such as professional insurance exams, regulatory standards, and real-world business data. INSEva comprises 38,704 carefully curated examples spanning various insurance sub-domains and difficulties. Model performance is evaluated using a combination of accuracy and domain-specific metrics tailored to each task.

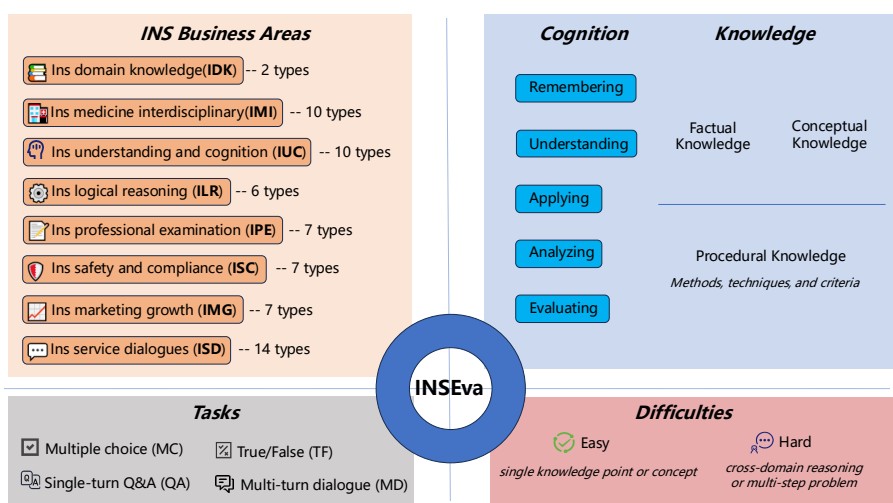

Figure 2: The taxonomy of INSEva benchmark.

## 3.2 TAXONOMY

To comprehensively assess models' capabilities in the insurance domain, we propose a multi-dimensional taxonomy for our evaluation benchmark. This taxonomy categorizes test items along four primary dimensions, including the business areas dimension, the task formats dimension, the difficulties dimension, and the cognition and knowledge dimension. The multi-dimensional taxonomy enables a fine-grained analysis of model performance across different aspects of insurance knowledge and applications.

### 3.2.1 DIMENSION 1: BUSINESS AREAS

We define a three-pillar, eight-dimension task taxonomy by integrating three sources of requirements: (i) the complementary strengths of general LLMs versus domain LLMs (language understanding/reasoning/safety alignment v.s. deep domain knowledge); (ii) the competency framework reflected in undergraduate and graduate insurance curricula (e.g., risk and actuarial science, insurance law and regulation, health/medical knowledge for health insurance, marketing and customer operations); and (iii) capability demands observed in end-to-end insurance applications (underwriting, claims, compliance, sales, and service). This leads to three capability pillars—Knowledge Literacy, Foundational General Skills, and Business-Integrated Skills—instantiated by eight business-area dimensions that jointly cover domain knowledge, cognition and reasoning, compliance, and workflow-centric dialogue and generation.

We categorize examples as follows: (1) **Insurance Domain Knowledge (IDK)** assesses professional knowledge literacy via two tasks—Insurance Knowledge Interpretation and Insurance Science; (2) **Insurance-Medical Interdisciplinary (IMI)** comprises ten cross-domain tasks spanning medical entity extraction and standardization, diagnosis/Q&A (including pet scenarios), and risk/prescription prediction; (3) **Insurance Understanding & Cognition (IUC)** includes ten information extraction and analysis tasks such as intent understanding, slot filling for products/insured objects/diseases, attribute extraction, clause interpretation, and liability/product selection analysis; (4) **Insurance Logical Reasoning (ILR)** emphasizes six quantitative and rule-based reasoning tasks, including actuarial science, financial mathematics, numerical computation, and exemption reasoning; (5) **Insurance Professional Examination (IPE)** simulate seven certification settings across insurance, physician/pharmacist/veterinarian practice, actuarial qualification, and salesperson licensing; (6) **Insurance Safety & Compliance (ISC)** defines seven tasks on information security, baseline controls, document compliance, value alignment, issue identification, fact-checking, and compliance verification; (7) **Insurance Marketing Growth (IMG)** evaluates seven tasks on customer segmentation, service summarization, marketing copy generation, recommendation script-

ing, investor education, population classification, and strategy formulation; (8) **Insurance Service Dialogue (ISD)** constructs fourteen multi-turn tasks covering the service lifecycle, including product/regulatory interpretation, underwriting and policy servicing, claims assessment and settlement, post-policy operations, planning/configuration, condition-based selection and comparison, and premium/benefit calculation. More details are provided in the Appendix.

### 3.2.2 DIMENSION 2: TASK FORMATS

According to the task format of the example, we classify examples into four types: (1) **Multiple choice** (MC): Tasks requiring selection of the correct option(s) from a set of alternatives; (2) **True/False** (TF): Binary judgment tasks requiring verification of statements' accuracy; (3) **Single-turn Q&A** (QA): Direct question-answer pairs requiring concise responses; (4) **Multi-turn dialogue** (MD): Extended conversations simulating real-world insurance consultations requiring contextual understanding and knowledge application across multiple exchanges.

### 3.2.3 DIMENSION 3: DIFFICULTIES

To assess the scalability of models across varying levels of expertise required for insurance operations, we categorize examples according to their difficulty: (1) **Easy**: Items can be answered directly without requiring deep understanding, or items only requiring the retrieval and application of a single knowledge point or concept; (2) **Hard**: Complex scenarios requiring integration and application of multiple knowledge elements, often involving cross-domain reasoning or multi-step problem-solving.

### 3.2.4 DIMENSION 4: COGNITION & KNOWLEDGE

To enable a granular assessment of models' cognitive processes and knowledge within the insurance domain, informed by established education assessment theory (Krathwohl, 2002), we categorize examples according to their alignment with specific cognitive dimensions and knowledge dimensions[1].

For the cognition dimension: (1) **Remembering** (Rem.): Recalling or recognizing basic insurance knowledge (e.g., facts, terminology) without deeper understanding; (2) **Understanding** (Und.): Comprehending insurance concepts through explanation, transformation, or inference to establish basic connections; (3) **Applying** (App.): Using acquired insurance knowledge or methods to solve problems in new contexts; (4) **Analyzing** (Ana.): Breaking down insurance information to clarify relationships, structures, and theories; (5) **Evaluating** (Eva.): Critically assessing the logic, structure, or effectiveness of insurance information and ideas.

For the knowledge dimension: (1) **Factual Knowledge** (FK): Basic elements of insurance (e.g., terminology, specific details); (2) **Conceptual Knowledge** (CK): Interrelationships among basic elements within insurance theory (e.g., classifications, principles); (3) **Procedural Knowledge** (PK): Methods, techniques, and criteria for determining when to use appropriate insurance procedures.

## 3.3 DATA CONSTRUCTION

This section details our systematic approach to developing a comprehensive benchmark in the insurance domain, encompassing data collection, data augmentation, and data quality control methods.

### 3.3.1 DATA COLLECTION

Our benchmark integrates data from diverse authoritative sources to ensure comprehensiveness and domain relevance. (1) We systematically extract historical examination questions, corresponding answers, and explanatory notes from professional Chinese certification platforms, subsequently restructuring these data to conform to the format; (2) We collect specialized terminology and regulatory standards from authoritative industry repositories, from which we extract core knowledge points and reformulate them into question-answer formats using LLM; (3) We curate internal business resources containing domain-specific knowledge and scenarios, transforming them into standardized

---

[1]It is worth noting that in the education assessment theory, there is also Creating under the cognition dimension and Metacognitive Knowledge under the knowledge dimension. Due to the seriousness and professionalism of the insurance domain, these two dimensions are not included in our taxonomy.

Table 1: Data statistics of different business areas and cognition and knowledge in INSEva. The absolute counts and relative percentage distribution are presented. The abbreviations are defined in Section 3.2.

| Business | Tasks | Len. | Num. | % |
|---|---|---|---|---|
| IDK | MC | 226 | 1336 | 3.5 |
| IMI | MC/TF | 372 | 9447 | 24.4 |
| IUC | MC | 402 | 7916 | 20.5 |
| ILR | MC/TF | 603 | 3923 | 10.1 |
| IPE | MC | 195 | 7932 | 20.5 |
| ISC | TF | 169 | 3417 | 8.1 |
| IMG | MC | 1141 | 1651 | 4.3 |
| ISD | QA/MD | 3230 | 3082 | 8.0 |

(a) Business Areas

| Type | Tags | Num. | % |
|---|---|---|---|
| Cognition | Rem. | 541 | 1.4 |
| | Und. | 8036 | 20.8 |
| | App. | 19339 | 50.0 |
| | Ana. | 7038 | 18.2 |
| | Eva. | 3750 | 9.7 |
| Knowledge | FK | 914 | 2.4 |
| | CK | 21047 | 54.4 |
| | PK | 16743 | 43.3 |

(b) Cognition and Knowledge

question-answer pairs using LLM. Finally, we collect about 40k examples from different sources. This multi-source approach ensures our benchmark encompasses both theoretical knowledge and practical applications within the insurance domain.

### 3.3.2 DATA AUGMENTATION

To enhance the robustness and validity of our benchmark, we implement several augmentation strategies for questions and answers on the collected data. (1) **Question Augmentation**: To promote linguistic diversity, we employ paraphrasing techniques to generate semantically equivalent questions with distinct syntactic expressions, thereby reducing potential model bias toward specific phrasings. Additionally, we make colloquial rewrites for standard questions, converting formal insurance terminology into colloquial expressions more representative of real-world customer interactions. This transformation enhances the benchmark's utility for customer-facing applications; (2) **Answer Augmentation**: We implement option randomization to mitigate selection bias (Zheng et al., 2024) for multiple-choice questions, ensuring uniform distribution of correct answers across options throughout the dataset. Finally, we get nearly 80k augmented examples in the benchmark.

### 3.3.3 DATA QUALITY CONTROL

To ensure the reliability and validity of our benchmark, we implement a multi-faceted quality control framework comprising rule-based, expert-based, and LLM-based validation modules. (1) **Rule-Based Module**: This module evaluates the quality of the samples through a series of rules. Specifically, we conduct: (1) character set validation to identify and rectify illegal characters, special symbols, or encoding anomalies, thereby ensuring Unicode compliance and eliminating corrupted elements; (2) structural validity verification for multiple-choice questions, confirming option completeness, absence of duplicates, and adherence to specified formatting requirements; and (3) content-specific quality checks, wherein samples are assessed against pre-defined criteria based on their inherent properties (e.g., questions about insurance products must explicitly reference the product name); (2) **Expert-Based Module**: This module ensures the faithfulness and consistency of ground truth in Q&A and dialogue samples through rigorous domain expert examination. A team of 10 insurance industry experts, averaging five years of experience and proficient in both Chinese and English, independently reviews the data to verify that ground truth responses are fully supported by and logically consistent with the provided input context. Any inconsistent or unsupported examples are removed. Inter-annotator agreement, measured by Cohen's Kappa, reaches 0.87, reflecting high consistency. Detailed annotation guidelines and tools are employed to ensure uniformity and reproducibility, as shown in Appendix E; (3) **LLM-Based Module**: To further enhance validation reliability and mitigate potential biases, we employ a multi-LLM voting approach. In this module, multiple distinct LLMs independently evaluate each sample. Final quality determinations are then made through aggregation of these independent assessments, leveraging the diverse perspectives of multiple models to achieve a more robust and objective validation process.

## 3.4 DATA STATISTICS

Finally, we attain a total of 38,704 examples, while the average length of the prompts of these examples is 905 tokens. Table 1a and Table 1b show more details of different business areas and different cognition and knowledge. A substantial proportion of questions in the benchmark belong to knowledge applying and procedural knowledge types, which provide a more realistic assessment of the model's practical capabilities in insurance.

## 3.5 EVALUATION METHODS

Due to the high risks of the insurance domain, we pay more attention to the correctness of the model's response in the insurance domain. Our evaluation framework employs different assessment approaches tailored to various question types in the benchmark.

For deterministic questions, such as multiple-choice and true/false questions, whether the model's response is the same as the ground-truth represents the correctness of the model's response. Specifically, we calculate *Accuracy* as the primary metric, representing the proportion of questions for which the model selects the correct option from the available choices. We further stratify performance analysis across different dimensions of our taxonomy to identify specific strengths and weaknesses in model capabilities.

For open-ended questions, including single-turn Q&A and multi-turn dialogues, we not only require that the model's response cover the ground truth as much as possible, but also require that no hallucinations appear in the response. Therefore, we propose two metrics, faithfulness and completeness, and implement an LLM-based evaluation pipeline for each metric that approximates human expert assessment. *Faithfulness* aims to measure the consistency of a model's response with the provided context, reflecting whether the answer contains hallucinations or unsupported claims. To calculate the faithfulness, we decompose the model's response into individual statements and determine whether the provided retrieval context substantiates each statement. Specifically, we use an LLM to decompose the model's response into a maximum of 20 statements. For each statement, the LLM determines whether it is supported by the provided retrieval context, assigning a binary score of 1 if supported and 0 otherwise. The overall faithfulness score is then calculated as the proportion of supported statements. *Completeness* aims to measure the extent to which a model's response covers the ground truth, reflecting whether the answer covers the valid information required. The core method involves decomposing the ground truth into statements and assessing whether each statement is present in the model's response. Specifically, we employ an LLM to decompose the ground truth into a maximum of 20 statements. The LLM then determines whether each statement is present within the model's response, assigning a binary score of 1 if present and 0 otherwise. The overall Completeness score is computed as the proportion of ground truth statements that are successfully recalled in the model's response.

## 4 EXPERIMENTS

## 4.1 EXPERIMENTAL SETTINGS

We select 9 state-of-the-art LLMs for evaluation, including closed-source, open-source, and domain-specific models, which are as follows: (1) **Closed-source:** GPT-4o (OpenAI, 2024); Doubao-1.5-pro-256k[2]; Gemini-2.5-pro[3]; (2) **Open-source:** Qwen2.5-72B-Instruct (Qwen et al., 2025); Qwen-QwQ-32B (Team, 2025); Deepseek-R1 (Guo et al., 2025); Qwen3-235B-A22B (Yang et al., 2025); (3) **Domain-specific:** Fin-R1 (Liu et al., 2025); DianJin-R1 (Zhu et al., 2025).

## 4.2 MAIN RESULTS

Based on pre-defined taxonomy and evaluation methods, we conduct a comprehensive assessment, and the results are presented in Table 2. A comprehensive evaluation of LLMs across different dimensions specific to the insurance domain reveals significant variations in overall performance.

---

[2]https://seed.bytedance.com/en/special/doubao_1_5_pro
[3]https://deepmind.google/models/gemini/pro/

Table 2: Main results for LLMs on INSEva across different business areas (abbreviations defined in Section 3.2).

| Models | IDK | IMI | IUC | ILR | IPE | ISC | IMG | ISD | | Avg. |
| | | | | | | | | *Faithful.* | *Complete.* | |
|---|---|---|---|---|---|---|---|---|---|---|
| GPT-4o | 84.24 | 79.25 | 84.90 | 56.68 | 74.83 | 73.84 | 94.85 | 82.26 | 80.97 | 79.09 |
| Doubao-1.5 | **90.17** | **80.88** | **85.44** | 58.75 | 87.11 | 78.35 | 94.93 | **86.86** | 82.17 | 82.74 |
| Gemini-2.5 | 88.14 | 80.68 | 84.54 | **78.27** | **87.75** | 79.13 | **95.75** | 75.54 | 85.24 | **83.89** |
| Qwen2.5 | 86.95 | 77.79 | 82.90 | 65.72 | 80.23 | **80.45** | 95.17 | 78.99 | 85.36 | 81.51 |
| Qwen-QwQ | 89.05 | 80.07 | 84.62 | 70.90 | 81.11 | 76.86 | 94.98 | 73.94 | **89.47** | 82.33 |
| Deepseek-R1 | 89.48 | 79.75 | 85.00 | 73.66 | 86.15 | 70.65 | 95.71 | 74.51 | 87.76 | 82.52 |
| Qwen3 | 86.48 | 79.51 | 84.62 | 71.74 | 85.01 | 73.34 | 95.29 | 72.87 | 88.40 | 81.92 |
| Fin-R1 | 80.34 | 72.48 | 76.40 | 61.90 | 67.20 | 74.56 | 87.41 | 70.70 | 81.04 | 74.67 |
| DianJin-R1 | 87.23 | 72.59 | 82.80 | 69.36 | 82.65 | 70.19 | 85.58 | 71.21 | 87.55 | 78.80 |

Gemini-2.5-pro demonstrates the highest aggregate performance, achieving a mean score of 83.89, while the domain-specific Fin-R1 model exhibits a comparatively lower mean score of 74.67. Notably, with the exception of the domain-specific model, the general LLMs (such as Qwen2.5, Deepseek-R1, etc.) maintain average scores above 80, indicating a relatively stable foundational capability in the insurance domain. However, a gap remains between current LLM performance and the level of expertise required to fully address complex real-world insurance problems.

Regarding granular dimensions, all evaluated models achieve excellent results in the Insurance Marketing Growth (IMG), with an average score of more than 94 points. The reason is that although IMG is composed of insurance data, the tasks in it are similar to general tasks in NLP (such as text summarization, etc.), and require the general language understanding and generation capabilities of LLMs. However, they generally perform poorly in Insurance Logical Reasoning (ILR), with scores generally lower than other dimensions, where the highest score is only 78.27 of Gemini-2.5-pro. This uneven distribution of capabilities reflects that the current LLMs still have room for improvement in complex reasoning tasks. Additionally, the generally low scores in the Insurance Safety Compliance (ISC) also expose the limitations of the model in dealing with strong regulatory scenarios. The mediocre performance of the models in these domain-specific tasks further emphasizes the need for insurance benchmarks to fill the gap in capability evaluation in the insurance domain.

Different LLMs show obvious differentiated capabilities. Gemini-2.5-pro achieves competitive performance in all dimensions, showing strong general performance. Qwen series LLMs have relative advantages in reasoning and dialogue generation. Fin-R1 and DianJin-R1 show typical domain-specific characteristics. They have excellent performance in the financial benchmark, but their performance in insurance is even worse than that of general LLMs. It shows that even though insurance is a subdomain of finance and there are many existing financial benchmarks, it is still very important to construct an insurance benchmark.

Through an in-depth analysis of the faithfulness and completeness under the insurance service dialogue (ISD), we can find that different types of LLMs show significant trade-off characteristics in these two key indicators. Specifically, reasoning models represented by Deepseek-R1 perform well in completeness, but their faithfulness index is relatively low. It shows that this type of model tends to generate more comprehensive answers through reasoning and knowledge integration, which can better cover the key information points in the ground truth, but at the same time it is also more likely to generate additional content that is not explicitly mentioned in the dataset. In contrast, the non-reasoning LLMs represented by GPT-4o show relatively high faithfulness, but the completeness is relatively low, reflecting that this type of model is more inclined to generate strict answers based on known information. This trade-off relationship between faithfulness and completeness has special practical significance in high-risk financial fields such as insurance. Insurance products usually involve complex clause details and strict regulatory requirements. False or inaccurate information generated by the model may cause users to make wrong insurance decisions, leading to serious economic losses or even legal disputes. Therefore, in practical applications, we believe that faithfulness should be given priority to ensure the reliability of model output.

Main results for LLMs across different cognition and knowledge are shown in Table 3. The experimental results reveal several noteworthy patterns across cognitive and knowledge dimensions. In terms of cognition dimensions, all models demonstrate stronger performance in lower-order cogni-

Table 3: Main results for LLMs across different cognition and knowledge.

| Models | Cognition | | | | | Knowledge | | | Avg. |
|---|---|---|---|---|---|---|---|---|---|
| | Rem. | Und. | App. | Ana. | Eva. | FK | CK | PK | |
| GPT-4o | 80.30 | 87.74 | 80.67 | 74.95 | 71.33 | 83.57 | 82.14 | 77.13 | 79.72 |
| Doubao-1.5 | 89.01 | 88.93 | 86.14 | 77.12 | 72.00 | 88.67 | **87.29** | 78.35 | 83.43 |
| Deepseek-R1 | **91.39** | **89.28** | **87.30** | 81.33 | 69.24 | **91.57** | 84.44 | 83.58 | **84.76** |
| Qwen3 | 88.23 | 87.81 | 85.89 | **81.71** | **75.17** | 88.69 | 84.82 | **83.59** | 84.48 |
| DianJin-R1 | 85.02 | 85.51 | 81.12 | 73.80 | 69.36 | 82.83 | 81.43 | 76.47 | 79.44 |

Table 4: Experimental results comparing insurance-specific model Finix-S1 with the base model.

| Models | IDK | IMI | IUC | ILR | IPE | ISC | IMG | ISD | | Avg. |
|---|---|---|---|---|---|---|---|---|---|---|
| | | | | | | | | Faithful. | Complete. | |
| Qwen-QwQ | 89.05 | 80.07 | 84.62 | 70.90 | 81.11 | 76.86 | **94.98** | 73.94 | **89.47** | 82.33 |
| Finix-S1 | **92.66** | **91.13** | **89.69** | **78.69** | 87.15 | **83.49** | 90.07 | **88.81** | 81.45 | **87.02** |
| Δ | +3.61 | +11.06 | +5.07 | +7.79 | +6.04 | +6.63 | -4.91 | +14.87 | -8.02 | +4.69 |

tive tasks such as Remembering (Rem.) and Understanding (Und.), with scores consistently above 85%, while showing relative weakness in higher-order cognitive tasks, particularly in Evaluation (Eva.) where scores drop below 75%. Notably, Deepseek-R1 exhibits exceptional performance in basic cognitive tasks (91.39% in Remembering), but experiences a significant performance degradation in evaluation tasks (69.24%), suggesting a common challenge in high-level cognition across all models. In terms of knowledge dimensions, models generally perform better in Factual Knowledge (FK) compared to Procedural Knowledge (PK), with Deepseek-R1 achieving the highest FK score of 91.57%. This pattern indicates that current models have a stronger grasp of basic facts and terminology than procedural operations in the insurance domain. The overall performance comparison shows Deepseek-R1 and Qwen3 leading with average scores of 84.76% and 84.48% respectively, significantly outperforming GPT-4o (79.72%) and DianJin-R1 (79.44%). This performance gap suggests that recent architectural improvements and training strategies have effectively enhanced models' capabilities across both cognition and knowledge dimensions.

### 4.3 INSURANCE-SPECIFIC MODEL TRAINING AND EVALUATION

To validate the effectiveness of the data construction pipeline, we also train a domain-specific model named **Finix-S1** based on Qwen-QwQ-32B model using reinforcement learning. As shown in Table 4, evaluation results demonstrate that Finix-S1 achieves a significant performance improvement. It surpasses the base model Qwen-QwQ by 4.69% on average, and current state-of-the-art general LLMs (e.g., Gemini) by over 3% on average. This advantage is particularly pronounced on the Insurance-Medical Interdisciplinary (IMI), which necessitates the integration of knowledge across multiple domains, with Finix-S1 exceeding the best model by more than 10%. This suggests that prevailing general LLMs exhibit limitations in cross-domain knowledge integration, highlighting the need for training paradigms that foster this capability. However, a key trade-off is observed: on tasks akin to general NLP (e.g., IMG), Finix-S1's performance lags behind that of general LLMs. This phenomenon, consistent across other domain-specific models, points to the challenge of catastrophic forgetting, where gains in specialized performance may come at the expense of general capabilities. More training and evaluation details are provided in the Appendix C.

## 5 CONCLUSION

We introduce INSEva, a comprehensive benchmark for the insurance domain featuring 38,704 high-quality examples. Our evaluations reveal that existing LLMs, including those specialized in finance, possess only basic competency and struggle with complex reasoning, often trading faithfulness for completeness. This highlights the unique challenges of the insurance sector. Ultimately, INSEva serves as a vital tool for practitioners in model selection and guides researchers in developing more robust architectures for high-risk domains.

## 6 ETHICS STATEMENT

This work adheres to the ICLR Code of Ethics. In this study, no human subjects or animal experimentation was involved. All datasets used were sourced in compliance with relevant usage guidelines, ensuring no violation of privacy. We have taken care to avoid any biases or discriminatory outcomes in our research process. No personally identifiable information was used, and no experiments were conducted that could raise privacy or security concerns. We are committed to maintaining transparency and integrity throughout the research process.

## 7 REPRODUCIBILITY STATEMENT

We are committed to fostering reproducible research. To this end, all components of our evaluation framework, including detailed descriptions of the methodology, evaluation scripts, and model interaction protocols, will be open-sourced upon publication. This will enable other researchers to apply our evaluation approach to their own models.

Regarding the dataset, the full benchmark contains sensitive commercial information and is proprietary, which prevents its public release. To balance the need for transparency with our confidentiality obligations, we will make a 10% stratified sample of the benchmark publicly accessible. This sample is carefully curated to reflect the distribution and complexity of the complete benchmark, providing a valuable resource for future research and analysis.

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

Table 5: Cross-Lingual evaluation results. The performance of several LLMs on a small constructed English benchmark is similar to that on Chinese.

| Models | IDK | IMI | IUC | ILR | IPE | ISC | IMG | ISD | | Avg. |
|---|---|---|---|---|---|---|---|---|---|---|
| | | | | | | | | Faithful. | Complete. | |
| Doubao-1.5 | **86.67** | **79.38** | 82.61 | 59.16 | 74.12 | 71.15 | 90.00 | 84.45 | 80.62 | 78.68 |
| Qwen2.5 | 84.59 | 75.85 | 81.62 | 64.73 | 68.50 | **92.82** | **94.14** | **85.51** | 71.15 | 79.88 |
| Qwen-QwQ | **86.67** | 77.57 | **83.24** | 70.82 | **75.29** | 81.66 | 93.09 | 70.48 | **83.26** | **80.23** |
| Qwen3 | 85.84 | 76.71 | 81.88 | **75.70** | 72.07 | 73.99 | 93.93 | 75.73 | 80.13 | 79.55 |
| Fin-R1 | 69.58 | 67.56 | 78.15 | 71.03 | 57.05 | 89.96 | 83.69 | 62.28 | 72.23 | 72.39 |

Fengbin Zhu, Wenqiang Lei, Youcheng Huang, Chao Wang, Shuo Zhang, Jiancheng Lv, Fuli Feng, and Tat-Seng Chua. Tat-qa: A question answering benchmark on a hybrid of tabular and textual content in finance, 2021. URL https://arxiv.org/abs/2105.07624.

Jie Zhu, Junhui Li, Yalong Wen, and Lifan Guo. Benchmarking large language models on cflue - a chinese financial language understanding evaluation dataset. In *Proceedings of the 62nd Annual Meeting of the Association for Computational Linguistics(ACL-2024)*, 2024.

Jie Zhu, Qian Chen, Huaixia Dou, Junhui Li, Lifan Guo, Feng Chen, and Chi Zhang. Dianjin-r1: Evaluating and enhancing financial reasoning in large language models. *arXiv preprint arXiv:2504.15716*, 2025.

# A USE OF LARGE LANGUAGE MODELS

During the preparation of this work, we used LLMs solely for the purpose of improving language and clarity. Specifically, LLMs were used for proofreading, grammar correction, and minor phrasing improvements.

We reviewed and edited all output generated by the LLMs, and take full responsibility for all content and ideas presented in this work.

# B EXPERIMENTS

## B.1 CROSS-LINGUAL EVALUATION

We conduct several cross-lingual experiments, as shown in Table 5. Specifically, we sample approximately 30% of the Chinese questions (around 10k) from our benchmark and translate them into English, and then test them across several models. Although the overall performance on English is slightly lower, the key conclusions remain consistent: performance is relatively low on ILR, a trade-off exists between faithfulness and completeness, and domain-specific models underperformed compared to general models. The results demonstrate that the benchmark is designed to assess the domain knowledge and reasoning abilities of LLMs in insurance, which remain consistent across languages without strong linguistic dependency.

## B.2 CORRELATION ANALYSIS

To validate the effectiveness of evaluation methods for open-ended questions, we select 98 instances from different categories in the INSEva benchmark and choose 4 different models, totally 392 responses for correlation analysis. We choose the BLEU and ROUGE family of methods as our baselines, which are widely used metrics for text quality assessment. Human evaluators are asked to rate the models' responses on faithfulness and completeness. The human-rated scores are considered as the optimal evaluation method, and we conduct a correlation analysis between various common automatic evaluation methods and the proposed evaluation method.

The correlation results are presented in Table 6. It can be observed that our evaluation method exhibits the best correlation, which means our evaluation method can achieve high consistency with human evaluation.

Table 6: Sample-level correlation of different metrics.

| Metrics | Faithful. | Complete. | Avg. |
|---------|-----------|-----------|------|
| Bleu-1 | 31.47 | -18.85 | 6.31 |
| Bleu-2 | 37.05 | -15.32 | 10.87 |
| Bleu-4 | 44.01 | -6.07 | 18.97 |
| Rouge-1 | 38.71 | -17.45 | 10.63 |
| Rouge-2 | 49.53 | -1.06 | 24.24 |
| Rouge-L | 43.86 | -7.63 | 18.12 |
| Ours | **73.60** | **55.78** | **64.69** |

Table 7: Experimental results for Finix-S1 with varying amounts of insurance-specific training data, while holding general data fixed.

| Models | IDK | IMI | IUC | ILR | IPE | ISC | IMG | ISD | | Avg. |
|--------|-----|-----|-----|-----|-----|-----|-----|-----|-----|------|
| | | | | | | | | *Faithful.* | *Complete.* | |
| Qwen-QwQ | 89.05 | 80.07 | 84.62 | 70.90 | 81.11 | 76.86 | **94.98** | 73.94 | **89.47** | 82.33 |
| 0% | 88.74 | 79.31 | 82.57 | 70.81 | 78.04 | 75.09 | 94.75 | 75.63 | 86.02 | 81.22 |
| 50% | 90.53 | 87.47 | 87.74 | 76.91 | 85.48 | 80.97 | 90.93 | 82.13 | 84.33 | 85.16 |
| 75% | 91.87 | 90.47 | 88.83 | 78.03 | 86.74 | 82.53 | 90.03 | 86.32 | 82.17 | 86.33 |
| 100% | **92.66** | **91.13** | **89.69** | **78.69** | 87.15 | **83.49** | 90.07 | **88.81** | 81.45 | **87.02** |

## C DETAILS OF FINIX-S1

Finix-S1's training consists of two stages, conducted on NVIDIA A100 GPUs. The model is first trained via Supervised Fine-Tuning (SFT) using 32 GPUs across 4 nodes. Then, we employ reinforcement learning to implement Group Relative Policy Optimization (GRPO), leveraging insurance Q&A data and incorporating a dual reward mechanism to improve both the accuracy of response formatting and content. In the RL stage, we perform 3 rollouts per sample, with a train batch size of 72.

The composition of our training data is as follows: (1) 108K general data samples collected from the GPQA dataset (Rein et al., 2023); (2) 347.3K samples constructed following our benchmark construction pipeline for SFT, along with an additional 36.5K samples utilized for reinforcement learning; (3) All training data are strictly separated from the evaluation data, ensuring that there is no data leakage between the training and evaluation sets.

We also evaluate the model by fixing the amount of general training data and varying the quantity of insurance-specific training data, as shown in Table 7. The results demonstrate that increasing the amount of insurance-specific training data consistently improves performance across multiple business areas. This indicates that current LLMs indeed lack sufficient insurance domain knowledge and the ability to apply knowledge. Furthermore, these results validate the effectiveness of our insurance data construction methodology. However, a slight performance decrease was observed specifically on the IMG. As discussed in the experimental section, this minor degradation aligns with the phenomenon of catastrophic forgetting, where increased specialization in the insurance domain leads to a decline in performance on certain more general tasks.

Here is an example. The user asks whether "hospitalization due to trauma within the past two years" can be covered by the Hao Yi Bao Long-term Medical Insurance (20-year edition). According to the standards, for first-time or non-continuous applicants, any illness or injury occurring before the inception of the insurance contract (categorized as a pre-existing condition, including trauma) is excluded from coverage. Finix-S1 with 0% insurance training data generates an erroneous response: "The system asserts that trauma is classified as accidental injury and, if the insurance is renewed continuously and no waiting period applies, coverage might be possible. It concludes with possible compensation after discussing different scenarios." While Finix-S1 with 100% insurance training data generates a correct response: "Hospitalization resulting from trauma within the past two years constitutes a pre-existing condition. According to the policy, this is categorically excluded from coverage (denied compensation)." The inadequately trained model demonstrates **a misunderstanding of domain-specific terminology**, specifically the distinction between "waiting period" and

"pre-existing condition" in insurance contracts. By conflating the rules for waiting periods—which primarily apply to post-contract incidents—with the distinct and absolute nature of pre-existing condition exclusions, the model erroneously infers that continuous renewal could eliminate the exclusion for prior trauma. This indicates a lack of nuanced comprehension of insurance lexicon and the relationships between contract clauses, leading to an incorrect compensation assessment for prior hospitalizations.

# D    DETAILS OF TAXONOMY

Our Insurance Business Areas dimensions consists of eight main areas:

1. Insurance Domain Knowledge (IDK):
    - Insurance Knowledge Interpretation
    - Insurance Science

2. Insurance-Medicine Interdisciplinary (IMI):
    - Medical Procedure Name Extraction
    - Medical Condition Name Extraction
    - Disease Relevance Discrimination
    - Insurance Hospital Name Standardization
    - Insurance Disease & Procedure Standardization
    - Pet Disease Diagnosis
    - Pet Disease Examination Q&A
    - Pet Medication Knowledge Q&A
    - Medical Prescription Disease Prediction
    - Disease Risk Prediction in Medical Reports

3. Insurance Understanding and Cognition (IUC):
    - Insurance Intent Understanding
    - Insurance Product Slot Recognition
    - Insured Object Slot Recognition
    - Insurance Disease Slot Recognition
    - Insurance Attribute Extraction
    - Insurance Clause Interpretation
    - Insurance Product Selection Analysis
    - Insurance Liability Analysis
    - Insurance Review Tag Recognition
    - Insurance Product Review Classification

4. Insurance Logical Reasoning (ILR):
    - Insurance Actuarial Science
    - Financial Mathematics
    - Financial Numerical Computation
    - Insurance Prior Exemption Reasoning
    - Insurance General Exemption Reasoning
    - Pure Outpatient Insurance Reasoning

5. Insurance Professional Examination (IPE):
    - Insurance Professional Qualification Exams
    - CICE Insurance Certification
    - Practicing Physician Qualification Exam
    - Practicing Pharmacist Qualification Exam
    - Practicing Veterinarian Qualification Exam
    - Chinese Actuary Examination

- Insurance Salesperson Certification Exam

6. Insurance Safety and Compliance (ISC):

   - Information Security
   - Security Baseline
   - Insurance Document Compliance
   - Insurance Value System
   - Insurance Issue Identification
   - Insurance Fact-Checking
   - Insurance Compliance Verification

7. Insurance Marketing Growth (IMG):

   - Insurance Target Population Positioning
   - Insurance Service Summary
   - Insurance Marketing Copy Generation
   - Insurance Product Recommendation Scripting
   - Insurance Investor Education Scripting
   - Insurance Population Identification & Classification
     - User Purchase Intention Classification
     - User Cognitive Level Classification
     - User Attitude Level
     - Insurance Type Preference
   - Insurance Service Strategy Formulation
     - Insurance Service Necessity Determination
     - Insurance Service Scenario Selection
     - Insurance Service Timing Decision
     - Insurance Service Scenario Expression

8. Insurance Service Dialogues (ISD):

   - Pre-Investment Consultation
     - Product Interpretation
     - Regulatory Information Interpretation
     - Underwriting-related
     - Policy-related
     - Platform-related
   - Post-Investment Issue Resolution
     - Claims Assessment-related
     - Claim Settlement-related
     - Post-policy Operations
   - Professional Allocation Advice
     - Planning and Configuration
     - Condition-based Product Selection
     - Product Recommendation
     - Insurance Type Comparison
     - Product Comparison
     - Premium and Benefit Calculation

This comprehensive taxonomy covers the full spectrum of insurance business areas, from fundamental domain knowledge to practical service applications, ensuring a thorough evaluation of AI models in the insurance domain.

保险营销增长-保险选配话术生成

回答正确case

## 问题1：
你是一个保险销售专家，我会给定一个潜在客户所咨询的保险选配问题，你需要从A、B、C、D四个选项中选出多个适合为该用户选配的选项。请注意，正确选项一定不止一个，不要进行单选。
最终结果只输出选项，不要输出额外信息，选项之间用英文逗号分隔。
潜在客户所咨询的保险选配问题是：【已婚有房贷买什么？】
选项是：
A.【定期寿险是重要保障，万一不幸身故，可以留一笔钱，确保家人维持日常开支。保额综合考虑贷款和年收入，建议保至退休年龄。重疾险同样可以考虑，确诊大病后一次性拿到一笔钱，可自由使用，用于弥补患病期间的收入损失、支付康复护理费。预算有限的情况下，建议选保额20/30年的产品，覆盖家庭责任最重的人生阶段。】
B.【推荐您考虑意外险，意外伤残身故后一次性拿到一笔钱，小意外可报销。价格便宜，保额高。您可以根据自身的需求和经济状况，考虑购买定期重疾险、医疗险、意外险等其他保险产品。在选择保险时，您需要考虑保险计划的保费和保额、保险计划的覆盖范围、是否有免赔额或限制、保险公司的声誉和客户评价、是否有附加服务，如紧急救援等。购买保险前，请仔细阅读合同的条款和细则。】
C.【作为已婚且有房贷的人，您肩负着照顾家人、偿还贷款的重担，压力较大。因此，建议您根据自身情况，合理配置保险。医疗险：优先保障，作为医保的补充，报销大额医药费。如果出现健康问题，可能无法购买。建议趁健康早购买保证续保的产品，锁定长期保障。】
D.【对于已婚且有房贷的人来说，最重要的是投资理财来增加收入，比如股票、基金或者房产投资。这样不仅能提高生活质量，还能更快还清贷款。当然，在投资之前，最好先了解一些金融知识，评估自己的风险承受能力，选择合适的投资渠道。】
输出：

## 回复1：
<think> 好，我现在需要帮用户解决保险选配的问题，题目是"已婚有房贷买什么？"，选项是A、B、C、D四个，要选多个正确选项。首先，我得仔细分析每个选项的内容，看看哪些适合已婚有房贷的人。首先看选项A。里面提到定期寿险和重疾险。定期寿险对于有房贷的人来说确实重要，因为万一身故，可以留一笔钱给家人还贷和维持生活。保额要考虑贷款和收入，保到退休年龄也合理。重疾险也是必要的，大病时的一次性赔付能覆盖收入损失和康复费用，尤其是预算有限时选20/30年期的，覆盖家庭责任重的阶段。这看起来很符合用户需求，所以A应该是正确选项之一。接下来是选项B，推荐意外险，并提到其他产品如定期重疾、医疗险、意外险。意外险确实价格低、保额高，能补充保障，尤其是意外伤残或身故的情况。但还提到购买前要考虑保费、保额、覆盖范围等，这部分比较通用，但意外险本身对有房贷家庭来说也是有用的，所以B可能也是正确选项之一。不过需要确认是否意外险是必须的，或者是否有其他更优先的选项。选项C强调医疗险，作为医保的补充，报销大额医药费，特别是保证续保的产品。已婚有房贷的人确实需要医疗险来应对高额医疗支出，避免因病返贫。而且趁健康时购买保更重要，所以C也是合理的选择。医疗险和A中的寿险、重疾险可以形成互补，覆盖不同风险。建议选投资理财，如股票、基金、房产来增加收入。虽然投资重要，但保险主要是风险管理，尤其是对于有负债的家庭，优先保障而非投资。D可能不太符合保险选配的问题，更多是理财建议，所以可能不正确。现在需要确认正确选项是否不止一个。A、B、C都提到了不同的保险类型，而D不相关。但问题是要选适合的保险选项，所以A、B、C中的哪些更合适？A中的定期寿险和重疾险都是核心保障，覆盖身故和大病风险，对房贷家庭很关键。B中的意外险也很必要，但可能优先级次于寿险和医疗险，不过B还提到其他产品如医疗险，但描述比较笼统。C明确推荐医疗险，并强调保证续保，这也是重要的。所以可能A、B、C都是正确选项？但需要再仔细看选项内容。选项B提到"推荐您考虑意外险"，然后后面是其他产品如定期重疾、医疗险、意外险，但可能的重点是意外险，而其他部分只是更一般性建议。而C则明确推荐医疗险，并给出具体建议。所以可能A和C更直接相关，而B的意外险虽然正确，但可能优先级较低，但题目要求选多个正确选项，所以可能A、B、C都正确，而D错误。不过需要再检查选项B是否真的正确。意外险确实适合，但问题是否必须选？比如，对于有房贷的家庭，意外险作为补充是有必要的，但可能不是最核心的。但题目可能认为B中的意外险建议是正确的，所以B也是正确选项。而B还提到其他医疗险，所以可能B中的保险建议是正确的，所以B应该选。综上，正确选项应该是A、B、C，而D不选。因为A覆盖寿险和重疾，B推荐意外险，C推荐医疗险，这些都是已婚有房贷人士需要的保障。D是投资建议，不属于保险选配，所以排除。 </think> A,B,C

*By deepseek-r1*

Figure 3: A case of correct answer in Chinese.

# E DETAILS OF HUMAN ANNOTATIONS

The human annotations in the domain expert module and correlation analysis consist of over 10 insurance industry experts with an average of 5 years of experience, based in China. They are proficient in both Chinese and English, belong to different organizations respectively, and can utilize their industry experience along with professional reference materials to analyze and evaluate answers to the questions.

Human experts have established annotation criteria that primarily evaluate answer quality from three dimensions - factuality, accuracy, and continuity - through granular assessment of knowledge points. The evaluation scale ranges from 5 (excellent) to 0 (poor):

- Factuality: Whether the response content adheres to factual truth and can be substantiated by evidence from retrieved knowledge sources.

- Accuracy: Whether the response demonstrates professional rigor without containing errors in knowledge, expertise, or factual information.

- Continuity: Whether the response maintains contextual coherence, enables smooth communication flow, and exhibits clear language expression through effective discourse transitions.

# F CASE STUDY

As illustrated in Figure 3 and Figure 4, DeepSeek-R1 leverages step-by-step deliberation to correctly select the life-critical triad of term-life, critical-illness and medical coverage for a mortgaged family, evidencing its strength in multi-hop causal reasoning under complex scenarios. In contrast, Figure 5 reveals that the non-reasoning Qwen2.5-72B-Instruct over-generalises the fine-grained intent "Does this fall under student insurance?" into a broad liability query (predicting D instead of B), yet precisely recognises a straightforward capacity question, underscoring its sensitivity to subtle semantic boundaries and the need for tailored prompting or post-calibration to mitigate misclassification.

Insurance marketing growth - generation of
insurance selection language

回答正确case/Answer the correct case

##Question 1:
You are an insurance sales expert, and I will provide a potential customer with an insurance selection question. You need to choose multiple options from A, B, C, and D that are suitable for the user's selection. Please note that there must be more than one correct option, do not make a single choice. The final result only outputs options, do not output additional information, and separate options with commas in English.
The insurance selection question that potential customers are inquiring about is: 'What should I buy with a mortgage if I am married?'
The options are:
A. Term life insurance is an important guarantee. In case of unfortunate death, you can leave a sum of money to ensure that your family can maintain their daily expenses. The insured amount should take into account both the loan and annual income, and it is recommended to insure until retirement age. Critical illness insurance can also be considered. After being diagnosed with a serious illness, one can receive a lump sum of money that can be freely used to compensate for income losses during the illness and pay for rehabilitation and nursing expenses. In the case of limited budget, it is recommended to choose products that guarantee 20/30 years and cover the life stages with the heaviest family responsibilities. 】
B. We recommend that you consider accident insurance, which allows you to receive a one-time payment after accidental disability or death, and can be reimbursed for minor accidents. Cheap price, high coverage. You can consider purchasing other insurance products such as periodic critical illness insurance, medical insurance, accident insurance, etc. based on your own needs and economic situation. When choosing insurance, you need to consider the premium and coverage of the insurance plan, the coverage of the insurance plan, whether there are deductibles or restrictions, the reputation and customer reviews of the insurance company, and whether there are additional services such as emergency rescue. Before purchasing insurance, please carefully read the terms and conditions of the contract. 】
C. As a married person with a mortgage, you bear the heavy burden of taking care of your family and repaying the loan, which puts a lot of pressure on you. Therefore, it is recommended that you allocate insurance reasonably according to your own situation. Medical insurance: priority coverage, as a supplement to medical insurance, reimburses large medical expenses. If there are health issues, it may not be possible to purchase. It is recommended to purchase products with guaranteed renewal while in good health, and lock in long-term protection. 】
D. For married people with mortgages, the most important thing is to invest and manage their finances to increase their income, such as investing in stocks, funds, or real estate. This not only improves the quality of life, but also helps to repay loans faster. Of course, before investing, it is best to first understand some financial knowledge, assess one's risk tolerance, and choose suitable investment channels. 】
Output:

##Reply 1:
Okay, I now need to help users solve the problem of insurance selection. The question is "What to buy with a mortgage when married?" There are four options: A, B, C, and D, and multiple correct options need to be selected. Firstly, I need to carefully analyze the content of each option to see which ones are suitable for married people with mortgages. Firstly, let's take a look at option A. It mentions term life insurance and critical illness insurance. Term life insurance is indeed important for people who have a mortgage, because in case of death, they can leave some money to repay the loan and maintain their livelihood for their family. The coverage amount should take into account both loans and income, and it is also reasonable to ensure coverage until retirement age. Critical illness insurance is also necessary, as a one-time payout in the event of a major illness can cover income loss and rehabilitation costs, especially when the budget is limited and a 20/30 year term is chosen to cover stages with heavy family responsibilities. This looks very much in line with our needs, so A should be one of the correct options. Next is option B, which recommends accident insurance and mentions other products such as periodic critical illness insurance, medical insurance, and accident insurance. Accident insurance does have a low price and high coverage, which can provide additional protection, especially in cases of accidental disability or death. But B also mentioned that before purchasing, one should consider factors such as insurance premium, coverage amount, etc. This part is quite common, but accident insurance itself is also useful for mortgage families, so B may also be one of the correct options. However, it needs to be confirmed whether accident insurance is necessary or if there are other priority options available. Option C emphasizes medical insurance as a supplement to medical insurance, reimbursing large medical expenses, especially for products that guarantee insurance renewal. Married people with mortgages do need medical insurance to cope with high medical expenses and avoid falling back into poverty due to illness. Moreover, it is important to purchase while healthy, so C is also a reasonable choice. Medical insurance and life insurance and critical illness insurance in A can complement each other and cover different risks. Option D suggests investing in financial management such as stocks, funds, and real estate to increase income. Although investment is important, insurance is primarily for risk management, especially for households with debt, prioritizing protection over investment. D may not be in line with the issue of insurance selection, but rather financial advice, so it may not be correct. Now we need to confirm if there is more than one correct option. A, B and C both mentioned different types of insurance, while D is not relevant. But the question is to choose the appropriate insurance option, so which of A, B, and C is more suitable? The term life insurance and critical illness insurance in A are both insurance and mentions other products such as periodic critical illness insurance, medical insurance, and accident insurance. Accident insurance does have a low price and high coverage, which can provide additional protection, especially in cases of death and serious illness, which are crucial for mortgage families. Although accident insurance in B is useful, its priority may be lower than life insurance and medical insurance. However, B also mentioned other products such as medical insurance, but the description is relatively vague. C explicitly recommends medical insurance and emphasizes the importance of ensuring renewal. So maybe A, B, and C are all correct options? But we need to carefully read the options again. Option B mentions 'I recommend you consider accident insurance ', followed by other products such as periodic critical illness insurance, medical insurance, and accident insurance. However, B's focus may be on accident insurance, while other parts are more like general recommendations. And C explicitly recommends medical insurance and provides specific recommendations. So it is possible that A and C are more directly related, and although the accident insurance in B is correct, its priority may be lower. However, the question requires selecting multiple correct options, so it is possible that A, B, and C are all correct, while D is incorrect. However, it is necessary to double check if option B is truly correct. Accident insurance is indeed suitable, but is it necessary to choose? For example, for families with mortgages, accident insurance as a supplement is necessary, but it may not be the most essential. But the question may assume that the accident insurance suggestion in B is correct, so B is also the correct option. And B also mentioned other products, but the user's issue is optional, so the accident insurance recommendation in B is correct, so B should choose it. Overall, the correct options should be A, B, C, and D should not be selected. Because A covers life insurance and critical illness, B recommends accident insurance, and C recommends medical insurance, all of which are necessary protections for married individuals with mortgages. D is investment advice and does not belong to insurance selection, so it is excluded.  </think> A,B,C
By deepseek-r1

Figure 4: A case of correct answer in English.

保险理解认知-保险意图理解/ Understanding Insurance Cognition -
Understanding Insurance Intent

回答错误case

## 问题1：
你是一名名专业的保险顾问，你可以对用户提问的意图进行理解
用户的提问是：我问的是这个属不属于学平险
以下哪个选项可以最恰当的描述用户的提问意图？
A.投保|查产品基础信息|查垫付方式
B.投保|查产品基础信息|查险种类型
C.操作类|扣费|未扣或扣费失败操作步骤
D.投保/产品规则咨询/保障哪些责任-小样本
你的输出只能是A、B、C、D四个字母中的一个

##Question 1:
You are a professional insurance consultant capable of understanding user intent.
The user asks: "I'm asking whether this belongs to student accident insurance."
Which option best describes the user's intent?
A. Insurance application | Check product basic information | Check advance payment method
B. Insurance application | Check product basic information | Check coverage type
C. Operational issue | Payment deduction | Steps to resolve failed or missing payment deductions
D. Insurance application/Product rule inquiry/Scope of coverage - Small sample
Your output must be a single letter among A, B, C, D.

## 回复1/ model answer1 :
D

## 正确答案1/ correct answer1 :
B

回答正确case

## 问题2：
你是一名专业的保险顾问，你可以对用户提问的意图进行理解
用户的提问是：能做20万的吗
以下哪个选项可以最恰当的描述用户的提问意图？
A.保全|续期续保|查产品续保规则|查续保年龄要求
B.保全|保单服务|保单变更|受益人变更
C.操作类|理赔|理赔材料
D.投保|查产品基础信息|查保额
你的输出只能是A、B、C、D四个字母中的一个

##Question 2:
You are a professional insurance consultant who can understand the intention of users' questions
The user's question is: Can we make it for 200000 yuan
Which of the following options best describes the user's intention to ask a question?
A. Preservation | Renewal | Check product renewal rules | Check renewal age requirements
B. Preservation | Policy Services | Policy Changes | Beneficiary Changes
C. Operational | Claims | Claims Materials
D. Insurance | Check product basic information | Check coverage amount
Your output can only be one of the four letters A, B, C, D

## 回复2/ model answer2 :
D

## 正确答案2/correct answer2:
D

By Qwen2.5-72B-Instruct

Figure 5: A case of wrong answer and correct answer.

Business: 保险理解认知 / Insurance Understanding and Cognition(IUC)
Task: 多项选择 / Multiple choice

Prompt:
## 背景：你是一名保险领域专家，以下是针对保险条款进行属性抽取的单项选择题，请直接给出正确答案的选项。
## Background: As an expert in the insurance sector, you are presented with a multiple-choice question concerning attribute extraction from insurance policy terms. Kindly select the correct option.

## 保险条款：
重大疾病住院前后门急诊医疗费用 指被保险人经医院诊断罹患重大疾病必须接受住院治疗，在住院前7日（含住院当日）和出院后30日（含出院当日）内，因与该次住院相同原因而接受重大疾病门急诊治疗时，被保险人需个人支付的、必需且合理的治疗重大疾病门急诊医疗费用（但不包括重大疾病特殊门诊医疗费用和重大疾病门诊手术医疗费用）。
## Insurance Terms:
Major illness-related pre-hospitalization and post-hospitalization outpatient and emergency medical expenses pertain to costs that the insured must bear when diagnosed with a critical illness and hospitalized. These costs apply for treatments received within 7 days prior to (including the day of admission) and 30 days following (including the day of discharge) hospitalization for the same medical reason, excluding expenses associated with special outpatient care and outpatient surgical treatments for major illnesses.

## 提取要求：
需要提取的属性是：【费用报销范围】；该属性的类型是：【ENUM】；该属性的取值范围是：【不限医保、医保外、医保内、医保内+医保外自费药、未知】
## Extraction Requirements:
The attribute to be extracted is [Coverage Range]; type: [ENUM]; value range: [Not restricted by insurance, Outside insurance coverage, Within insurance coverage, Within insurance + self-paid medication outside insurance, Unknown]

## 选项：
A.医保内+医保外自费药
B.医保内
C.医保外
D.不限医保
## Options:
A.    Within insurance + self-paid medication outside insurance
B.    Within insurance coverage
C.    Outside insurance coverage
D.    Not restricted by insurance

Answer:
D

Figure 6: A multiple-choice example.

Business:保险安全合规 / Insurance Safety and Compliance (ISC)
Task: 判断 / True-False

Prompt
## 背景：你是一名保险文案合规性审核专家，你可以识别一段保险文案是否合规，如果合规，则回答"是"，如果不合规，则回答"否"。
## Background: As a compliance review expert for insurance copy, you have the ability to determine its compliance status. Respond with 'Yes' if the copy is compliant, or 'No' if it is not.

## 不合规情况：
1.出现歧视行为、特殊群体刻板印象词汇
2.夸大产品保障
3.宣传产品限时限量，刺激购买
4.引用不真实、不准确的数据和资料
5.使用政府、监管等描述进行不当背书
6.使用红包补贴、免费宣传进行诱导销售
7.涉及法律功能不实描述
8.使用绝对化用语类、极限词
9.隐瞒限制条件
## Non-Compliance Assessment:
1. Discriminatory language or stereotypes about specific groups
2. Overstating the coverage provided by the product
3. Creating urgency by promoting time-limited or quantity-limited offers
4. Quoting inaccurate or misleading data and information
5. Inappropriately using endorsements related to government or regulatory authorities
6. Engaging in sales inducement through offers like discounts or free incentives
7. Misrepresenting legal functionalities
8. Using absolute or extreme language
9. Failing to disclose any limitations or conditions

## 保险文案：
随着生活节奏的加快，很多人对于长期保险有了新的认识和需求。尤其是上班族，面对日益增长的医疗费用，如何加强自身的保障，成为他们最关心的问题。好医保·长期医疗（0免赔），为你提供全方位的医疗保障。每月保费11.77元起就可以享受最高400万的医疗保障。日常小病小痛住院能报销，重大疾病产生的住院医疗费用能报销。社保外的进口药、自费药、靶向药以及120万的car-t抗癌针都可以报销。不管你是年轻人还是老年人，好医保·长期医疗（0免赔）都可以为你提供最全面的医疗保障。现在就来投保，让你的生活更加安心！
## Insurance Copy
As fast-paced living becomes the norm, there 's a growing awareness and need for long-term medical insurance. Office workers, in particular, are increasingly concerned about rising medical costs and seek to enhance their coverage. The Good Health Insurance · Long-term Medical plan (with zero deductible) offers comprehensive medical protection. With premiums starting as low as 11.77 yuan per month, you can receive up to 4 million yuan in medical coverage. From minor illnesses to significant diseases, hospitalization expenses can be reimbursed. Coverage includes imported, self-funded, and targeted medications, as well as up to 1.2 million yuan for CAR-T cancer treatment not covered by social insurance. Whether young or old, Good Health Insurance · Long-term Medical (zero deductible) offers you the broadest medical coverage available. Sign up now for a more secure future!

Answer
否
False

Figure 7: A True/False example.

# G  EXAMPLES

To illustrate the heterogeneous nature of our evaluation framework, we showcase representative examples across three distinct task formats: a multiple-choice example as demonstrated in Figure 6, a True/False example as shown in Figure 7, a Q&A example as presented in Figure 8, and a multi-turn dialogue example as presented in Figure 9.

Business: 保险服销对话/ Insurance Service Dialogue
Task: 单轮问答/Single-turn Q&A

Prompt:
## 背景: 你是一名专业的保险规划师，针对问题根据检索内容给出最恰当的回答。
## Background: As a professional insurance planner, your role is to offer the most suitable response to inquiries based on the information retrieved.

## 问题: 灵活宝·年金对高风险职业的人有什么特殊规定吗？
## Query: Are there specific terms or conditions in the Flexible Treasure Annuity plan that apply to individuals working in high-risk jobs?

## 检索信息：
检索1: 灵活宝·年金这款产品在投保前无需特殊体检，本产品没有健康告知。能否投保成功以保险公司审核为准。
检索2: 职业类别指保险公司根据危险程度，给不同职业划分的风险级别。且通常会限制高危职业人群投保。一般来说，白领、教师、推销员等工作风险较小的人群为1-3类；部件装配工人、鱼贩肉贩等工作中稍有风险的人群为4类；冲压工、车床工等工作风险较大的人群为5类；6类职业往往是高空作业人群、消防员等工作风险极高的人群。###职业类别没有统一标准，比如中式厨师，有的保险公司划分到2类，有的属于4类，不同公司有自己的标准，请以具体投保的产品为准。 ###你可以在投保页的《投保须知》或《保险条款》中查阅具体的职业分类。如未找到相关入口，可点击【立即投保】，页面底部会有条款链接。
检索3: 灵活宝·年金未规定职业要求。 ### 小财提示: 本产品被保人年龄范围须为18-65周岁且仅支持投保人为本人投保。
## Retrieved Information:
Retrieval 1: The Flexible Treasure Annuity plan does not require a medical examination or health history disclosure before enrollment. Approval is subject to the insurance company's discretion.
Retrieval 2: Occupation categories are risk levels assigned by insurers based on job-related hazards. High-risk professions usually face restrictions in policy acquisition. Low-risk jobs like those of white-collar workers, teachers, and sales agents fall into categories 1-3; moderate risk jobs, such as assembly line workers and fish/meat vendors, are category 4; higher-risk jobs like those of stamping and lathe operators are category 5; category 6, high-risk jobs include those performed at great heights and firefighting. ### Occupational classifications are not standardized across companies, for example, a Chinese chef could be considered category 2 by one company and category 4 by another. It's crucial to check the standards of the specific policy. ### The <Insurance Instructions> or <Policy Terms> on the application page list occupation classifications. If inaccessible, click [Purchase Now] for links to these terms at the bottom.
Retrieval 3: Flexible Treasure Annuity does not impose occupational restrictions. ### Note: Eligible ages for policyholders are 18-65 years old, and the policy can only be purchased by the insured individual themselves.

Answer:
针对高风险职业人群，没有特殊规定。
There are no specific rules or conditions applied to individuals in high-risk jobs.

Figure 8: A single-turn Q&A example.

Business: 保险服销对话/ Ins service dialogues (ISD)
Task: 多轮对话/ Multi-turn dialogue (MD)

Prompt:
# background: 你是一名专业的保险规划师，你需要根据对话上文以及检索内容给出最恰当的回答。/You are a professional insurance planner. You need to provide the most appropriate response based on the conversation history and retrieved information.

## Context:
Q: 你好呀 /Hello there
A: 您好，遇到了什么问题了吗？有什么可以帮您的？ /Hello, have you encountered any issues? How can I assist you?
Q: 这样，我想买个意外险，你们有啥推荐吗？ /Well, I want to purchase accident insurance. Do you have any recommendations?
A: 有的，我们卖的比较好的有中国平安综合意外险(2024版)，您可以看看这个 /Yes, one of our best-selling products is Ping An of China Comprehensive Accident Insurance (2024 Edition). You may review this option.
Q: 哦哦，好的，我看看，发现我买不了，投保失败，这个怎么回事？ /Okay, let me check. It shows my application failed. Why is this happening?
A: 您今年多大了，身体状态怎么样？ /May I ask your age and current health status?

## Query: 我今年58岁，身体健康状况良好，我无法完成投保流程，这是什么原因呢？ / I am 58 years old with good health, but I cannot complete the insurance application. What is the reason?

## Retrieve:
检索1: 若您无法投保中国平安综合意外险(2024版)这款产品，可能是因为：1. 被保人不满足：18-55周岁，具有完全民事行为能力和完全生活自理能力的自然人。2. 被保人的职业不属于1-3类人群。 ### 小宝提示: 若仍以上问题，也可拨打平安产险官方客服电话95511-5进行咨询。
检索2: 投保时遇到失败情况，可能有以下几点原因：1、年龄不符；2、未通过健康告知；3、从事保险公司限制的高危职业。若因为以上原因不能投保，建议选择其他保险产品，若投保失败原因不明，可以直接拨打承保公司客服电话进行询问。
检索3: 中国平安综合意外险(2024版)仅承保教师、学生、白领等1-3类职业，对于4类及以上职业人员，以及1-3类职业人员因4类及以上职业行为出险的，保险公司不承担赔偿责任。 ### 小宝提示: 你可以点击【我要投保】，在【职业告知】中查看具体的职业等级划分。
Retrieval 1:If you cannot apply for Ping An of China Comprehensive Accident Insurance (2024 Edition), possible reasons include:
1.Insured party criteria unmet: Age 18-55, a natural person with full civil capacity and independent living ability. 2.Occupation not classified as Category 1-3. Xiaobao Tip: If the issue persists, contact Ping An Property & Casualty Insurance via official hotline 95511-5.
Retrieval 2:Common reasons for policy application failure: 1.Age mismatch 2.Failed health declaration 3.Engagement in high-risk occupations restricted by insurers. If ineligible, consider alternative products. For unresolved issues, contact the insurer directly.
Retrieval 3:Ping An of China Comprehensive Accident Insurance (2024 Edition) only covers Category 1-3 occupations (e.g., teachers, students, office workers). Claims arising from Category 4+ occupations or occupational activities are excluded. Xiaobao Tip: Check occupational classifications in the "Occupation Declaration" section during application.

Answer:
根据中国平安综合意外险(2024版)的产品规定，最大的投保年龄为55岁，因此您目前的年龄超出了该产品的投保年龄限制，所以无法完成投保。建议您可以考虑其他适合老年人的意外险产品来获得相应的保障。 / According to the policy terms of Ping An of China Comprehensive Accident Insurance (2024 Edition), the maximum insurable age is 55 years old. Your current age exceeds this limit, resulting in application failure. We recommend considering other accident insurance products suitable for seniors to obtain appropriate coverage.

Figure 9: A multi-turn dialogue example.

数据获取 /Data Collection

## 任务：
将以下提供的保险领域文本内容，转化为结构化的问答（Q&A）形式。
```
{知识点}
```

## 要求：
1. **提取关键信息：** 准确识别并提取文本中关于保险概念、定义、流程、特点、条款或案例等的关键知识点。
2. **转化为问答对：** 将每个关键知识点设计为一个清晰的问题和相应的答案。
3. **确保信息准确性：** 问答内容必须忠实于原文，不得增加、删除或曲解原文提供的信息。
4. **问题明确易懂：** 设计的问题应直接指向某个知识点，语言简洁，易于非专业人士理解（如果适用）。
5. **答案简洁明了：** 答案应直接回应问题，内容精炼，提取原文的核心解释。
6. **覆盖主要内容：** 确保文本中的主要知识点都得到了覆盖，形成相应的问答。
7. **格式规范：** 以清晰的问答列表形式呈现，例如使用"问题："和"答案："作为每一对问答的标记。

## 输出规范：
以json格式输出：{"question":"问题", "answer":"答案"}

## 待处理文本：
```
{text}
```

## Task:
Convert the following provided text content in the insurance field into a structured question-and-answer (Q&A) format.
```
{knowledge points}
```

## Requirements:
1.Extract key information: Accurately identify and extract key knowledge points in the text regarding insurance concepts, definitions, processes, characteristics, clauses, or cases, etc.
2.Transform into question-and-answer pairs: Design each key knowledge point into a clear question and corresponding answer.
3.Ensure information accuracy: The Q&A content must be faithful to the original text and shall not add, delete, or misinterpret the information provided in the original text.
4.Questions clear and easy to understand: The designed questions should directly point to a certain knowledge point, with concise language and be easy for non-professionals to understand (if applicable).
5.Answers concise and clear: Answers should directly respond to the questions, with refined content and extract the core explanations from the original text.
6.Cover main content: Ensure that the main knowledge points in the text are all covered to form corresponding Q&As.
7.Format specification: Present in a clear Q&A list format, for example, using "Question:" and "Answer:" as the markers for each Q&A pair.

## Output specification:
Output in json format: {"question":"Question", "answer":"Answer"}

## Text to be processed:
```
{text}
```

Figure 10: The prompt for converting knowledge to questions.

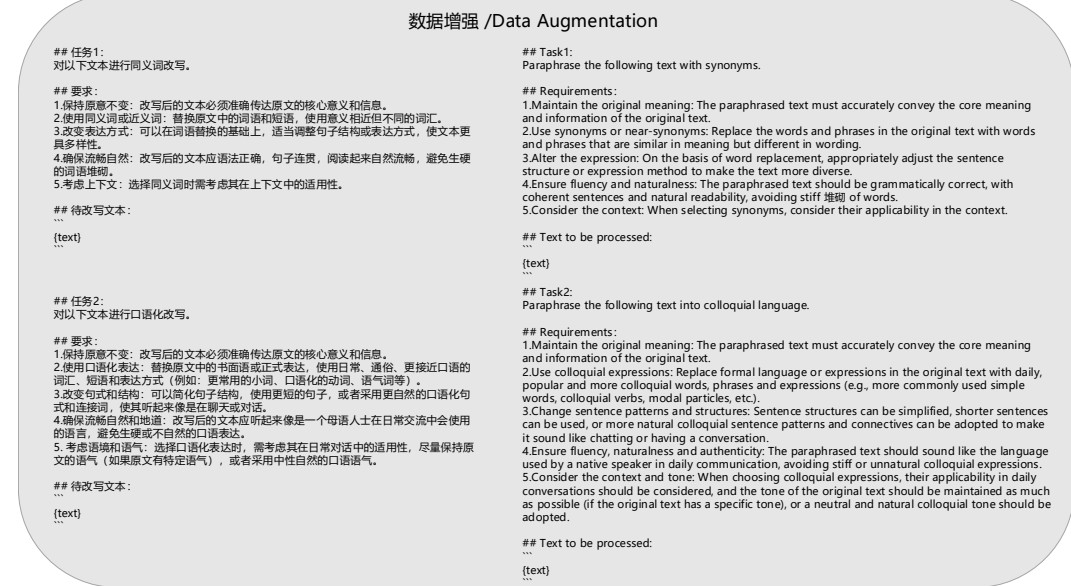

数据增强 /Data Augmentation

## 任务1：
对以下文本进行同义词改写。

## 要求：
1.保持原意不变：改写后的文本必须准确传达原文的核心意义和信息。
2.使用同义词或近义词：替换原文中的词语和短语，使用意义相近但不同的词汇。
3.改变表达方式：可以在词语替换的基础上，适当调整句子结构或表达方式，使文本更具多样性。
4.确保流畅自然：改写后的文本应语法正确，句子连贯，阅读起来自然流畅，避免生硬的词语堆砌。
5.考虑上下文：选择同义词时需考虑其在上下文中的适用性。

## 待改写文本：
```
{text}
```

## 任务2：
对以下文本进行口语化改写。

## 要求：
1.保持原意不变：改写后的文本必须准确传达原文的核心意义和信息。
2.使用口语化表达：替换原文中的书面语或正式表达，使用日常、通俗、更接近口语的词汇、短语和表达方式（例如：更常用的小词、口语化的动词、语气词等）。
3.改变句式和结构：可以简化句子结构，使用更短的句子，或者采用更自然的口语化句式和连接词，使其更听起来像是在聊天或对话。
4.确保流畅自然和地道：改写后的文本应听起来像是一个母语人士在日常交流中会使用的语言，避免生硬或不自然的口语表达。
5.考虑语境和语气：选择口语化表达时，需考虑其在日常对话中的适用性，尽量保持原文的语气（如果原文有特定语气），或者采用中性自然的口语语气。

## 待改写文本：
```
{text}
```

## Task1:
Paraphrase the following text with synonyms.

## Requirements:
1.Maintain the original meaning: The paraphrased text must accurately convey the core meaning and information of the original text.
2.Use synonyms or near-synonyms: Replace the words and phrases in the original text with words and phrases that are similar in meaning but different in wording.
3.Alter the expression: On the basis of word replacement, appropriately adjust the sentence structure or expression method to make the text more diverse.
4.Ensure fluency and naturalness: The paraphrased text should be grammatically correct, with coherent sentences and natural readability, avoiding stiff 堆砌 of words.
5.Consider the context: When selecting synonyms, consider their applicability in the context.

## Text to be processed:
```
{text}
```

## Task2:
Paraphrase the following text into colloquial language.

## Requirements:
1.Maintain the original meaning: The paraphrased text must accurately convey the core meaning and information of the original text.
2.Use colloquial expressions: Replace formal language or expressions in the original text with daily, popular and more colloquial words, phrases and expressions (e.g., more commonly used simple words, colloquial verbs, modal particles, etc.).
3.Change sentence patterns and structures: Sentence structures can be simplified, shorter sentences can be used, or more natural colloquial sentence patterns and connectives can be adopted to make it sound like chatting or having a conversation.
4.Ensure fluency, naturalness and authenticity: The paraphrased text should sound like the language used by a native speaker in daily communication, avoiding stiff or unnatural colloquial expressions.
5.Consider the context and tone: When choosing colloquial expressions, their applicability in daily conversations should be considered, and the tone of the original text should be maintained as much as possible (if the original text has a specific tone), or a neutral and natural colloquial tone should be adopted.

## Text to be processed:
```
{text}
```

Figure 11: The prompt for data augmentation.

# H  PROMPTS

## H.1  PROMPTS FOR DATA CONSTRUCTION

Figure 10 shows the prompt for converting knowledge to questions. Figure 11 shows the prompt for data augmentation. Figure 12 shows the prompt for data quality control.

## H.2  PROMPTS FOR EVALUATION

Figure 13 and Figure 14 shows the prompt for answer content splitting and matching for calculating faithfulness.

数据质量控制 /Data Quality Control

判断相关性
## 任务：
根据输入的文本内容，判断内容其是否与保险领域
相关。若相关，则返回 true；若不相关，则返回
false。

## 输出规范：
以json格式输出:{"result":"true或者false"}

## 输入文本：
{text}

判断常识性
## 任务：
判断输入文本是否包含常识性错误。如果文本中存
在与普遍认可的事实或常识相悖的内容，则返回
true；否则返回false。

## 输出规范：
以json格式输出:{"result":"true或者false"}

## 输入文本：
{text}

Judgment of Relevance
## Task1:
Based on the input text content, determine whether the content is relevant to the insurance domain. If relevant, return true; if not relevant, return false.

## Output Specification:
Output in JSON format: {"result": "true or false"}

## Input Text:
{text}

Judging Common Sense
## Task2:
Determine whether the input text contains common sense errors. If the text contains content that contradicts generally accepted facts or common sense, return true; otherwise, return false.

## Output Specification:
Output in JSON format: {"result": "true or false"}

## Input Text:
{text}

Figure 12: The prompt for data quality control.

## 任务描述
给你一个question和answer，请结合question信息，并分析answer文本的复杂度，将answer的文本分解成最多20个可以完全可理解的陈述句，确保任何陈述中不使用代词。请直接输出最终结果，除此以外不要输出其它内容。
## Task Definition
Given a question and its answer, use the information from the question to analyze the answer's complexity. Decompose the answer into no more than 20 clear and understandable declarative sentences, making sure not to use pronouns in any of the statements. Output only the final result without any additional content.

输入：
Input：
{
    "question" : {text},
    "answer": {text}
}

## 参考案例
输入：
{
    "question":"平安中老年人意外保险特定区域和其他区域的核心差异有哪些？",
    "answer":"您好，平安中老年人意外保险的特定区域包括公共交通工具内部和公益性文化设施内，其他区域则是这些地方之外的普通区域哦。在特定区域内发生意外事故，不同套餐的保额会更高，比如升级版在特定区域身故残疾保额能到20万，其他区域是10万。另外，意外医疗报销的免赔额是100元，经社保报销后按80%赔付，未经社保的按60%赔付，这点在两类区域都适用。"
}
输出：
["平安中老年人意外保险的特定区域包括公共交通工具内部和公益性文化设施内","平安中老年人意外保险的其他区域是特定区域之外的普通区域。", "在特定区域内发生意外事故时，不同套餐的保额会更高。", "升级版在特定区域身故残疾保额能达到20万。", "升级版在其他区域身故残疾保额为10万。", "意外医疗报销的免赔额是100元。", "经社保报销后，意外医疗按80%赔付。", "未经社保报销时，意外医疗按60%赔付。", "意外医疗报销规则在特定区域和其他区域都适用。"]
## Reference Example
Input：
{
    "question":" What are the primary differences between designated areas and other areas in Ping An Senior Accident Insurance? ",
    "answer":" Hello, Ping An Senior Accident Insurance defines specific areas as the interiors of public transportation vehicles and public cultural facilities. Other areas denote regular regions outside these specific zones. Coverage amounts vary, with higher benefits for accidents occurring within designated areas as per different plans. For instance, the upgraded plan provides death and disability coverage of up to 200,000 in designated areas, whereas it offers 100,000 in regular areas. Moreover, the medical reimbursement for accidents has a deductible of 100 yuan. Reimbursements are made at 80% post-social insurance coverage and at 60% when not covered by social insurance. These conditions apply across both designated and other areas. "
}
Output:
["Specific areas for Ping An Senior Accident Insurance include the interiors of public transportation and public cultural facilities.", "Other areas for Ping An Senior Accident Insurance are regular zones outside these specific areas.", "Coverage is higher for accidents occurring in specific zones based on the plan chosen.", "The upgraded plan provides up to 200,000 in death and disability coverage for specific areas.", "The upgraded plan provides 100,000 in death and disability coverage in other areas.", "A 100 yuan deductible applies to accident medical reimbursements.", "Accident medical claims are covered at 80% after social insurance reimbursement.", "Accident medical claims are covered at 60% without social insurance reimbursement.", "The medical reimbursement policy applies to both specific and regular areas."]

Figure 13: The prompt for answer content splitting.

## 任务描述
你的任务是根据给定的context判断statements中所有陈述句的忠诚度。对于每个陈述句，如果该陈述句可以直接基于context推理出来或者检索出来，则返回判定为1；如果该陈述句不能直接基于context推理或者检索出来，则返回判定为0。请直接输出结果，除此以外不要输出其它内容。

## Task Description
Your task is to assess the fidelity of statements based on the given context. For each statement, assign a value of 1 if it can be directly inferred or sourced from the context, or 0 if it cannot. Please output only the results without any additional content.

输入：
Input:
{
    "context":{text},
    "statements": {text}
}

## 参考案例
输入：
{
    "context":"检索1:线上投保通常可以选择多种便捷的付款方式，包括银行卡支付（借记卡、信用卡）、第三方支付平台（如支付宝、微信支付）、以及保险公司的官方APP或网站绑定的支付渠道。一些保险公司还支持分期支付以减轻一次性付款压力。选择付款方式时要确保符合保险公司要求，并注意保费扣款时间，以避免因支付失败导致保单生效受影响。此外，还需注意保存付款凭证以备后续查询或理赔使用。\n检索2:保费可采取多种支付方式，通常包括现金支付、银行卡转账支付、支付宝支付等。",
    "statements": ["确保信用卡在可用方式列表中。","综上，信用卡支付是可行的。","但需结合具体投保渠道的设置。"]
}
输出：
[
    {{"statement":"确保信用卡在可用方式列表中。", "verdict": 1}},
    {{"statement":"综上，信用卡支付是可行的。", "verdict": 1}},
    {{"statement":"但需结合具体投保渠道的设置。", "verdict": 1}}
]

## Reference Example
Input:
{
    "context":" Source 1: Online insurance enrollment typically allows for various easy payment options, such as bank card payments (both debit and credit cards), third-party platforms (like Alipay, WeChat Pay), and payment channels linked to the insurance company's app or website. Installment plans are also available with some insurers to alleviate the pressure of full upfront payment. Ensure compliance with insurer payment guidelines and be mindful of payment schedules to prevent issues with policy activation due to payment hiccups. It's also important to keep the payment receipt handy for future inquiries or claims.\nSource 2: Premiums can be settled through multiple payment methods, commonly including cash, bank card transfers, and Alipay. ",
    "statements": ["Make sure that credit card payment is an option listed among the available methods.", "In conclusion, using a credit card for payment is possible.","Nonetheless, it should align with the specific arrangements of the insurance purchase platform."]
}
Output:
[
    {"statement":"Make sure that credit card payment is an option listed among the available methods.", "verdict": 1},
    {"statement":" In conclusion, using a credit card for payment is possible.", "verdict": 1},
    {"statement":" Nonetheless, it should align with the specific arrangements of the insurance purchase platform.", "verdict": 1}
]

Figure 14: The prompt for answer matching.

## 任务描述
给你一个question和answer，请结合question信息，并分析answer文本的复杂度，将answer的文本分解成最多20个可以完全可理解的陈述句，确保任何陈述句中不使用代词。陈述句的内容必须是answer的原文内容，不得进行任何隐含的数学计算或推导。请直接输出最终结果，除此以外不要输出其它内容。
输入：
{
    "question": "{}",
    "answer": {}
}

## 参考案例
输入：
{
    "question":"平安中老年人意外保险特定区域和其他区域的核心差异有哪些？",
    "answer":"您好，平安中老年人意外保险的特定区域包括公共交通工具内部和公益性文化设施内，其他区域则是这些地方之外的普通区域域。在特定区域内发生意外事故，不同套餐的保额会更高，比如升级版在特定区域身故残疾保额能到20万，其他区域10万。另外，意外医疗报销的免赔额是100元，经社保报销后按80%赔付，未经社保的按60%赔付，这点在两类区域都适用。"
}
输出：
["平安中老年人意外保险的特定区域包括公共交通工具内部和公益性文化设施内。","平安中老年人意外保险的其他区域是特定区域之外的普通区域。","在特定区域内发生意外事故时，不同套餐的保额会更高。","升级版在特定区域身故残疾保额能达到20万。","升级版在其他区域身故残疾保额为10万。","意外医疗报销的免赔额是100元。","经社保报销后，意外医疗按80%赔付。","未经社保报销时，意外医疗按60%赔付。","意外医疗报销规则在特定区域和其他区域都适用。"]

## Task Description
You are given a question and an answer. Please analyze the complexity of the answer text in combination with the question information, and decompose the answer text into a maximum of 20 fully understandable declarative sentences. Ensure that no pronouns are used in any statement. The content of the declarative sentences must be the original content of the answer, and no implicit mathematical calculations or derivations should be performed. Please directly output the final result, and do not output anything else.
Input:
{
    "question": "{}",
    "answer": {}
}

## Reference Case
Input:
{
    "question": "What are the core differences between the specific areas and other areas of Ping An Middle-aged and Elderly Accident Insurance?",
    "answer": "Hello, the specific areas of Ping An Middle-aged and Elderly Accident Insurance include the interiors of public transport vehicles and public welfare cultural facilities, and other areas are ordinary areas outside these places. In the event of an accident in a specific area, the insurance amount of different packages will be higher. For example, the death and disability insurance amount of the upgraded package can reach 200,000 yuan in specific areas, and 100,000 yuan in other areas. In addition, the deductible for accident medical reimbursement is 100 yuan. After reimbursement through social security, it is compensated at 80%, and if not reimbursed through social security, it is compensated at 60%. This applies to both types of areas."
}
Output:
["The specific areas of Ping An Middle-aged and Elderly Accident Insurance include the interiors of public transport vehicles and public welfare cultural facilities.", "The other areas of Ping An Middle-aged and Elderly Accident Insurance are ordinary areas outside the specific areas.", "When an accident occurs in a specific area, the insurance amount of different packages will be higher.", "The death and disability insurance amount of the upgraded package can reach 200,000 yuan in specific areas.", "The death and disability insurance amount of the upgraded package is 100,000 yuan in other areas.", "The deductible for accident medical reimbursement is 100 yuan.", "After social security reimbursement, accident medical expenses are compensated at 80%.", "When not reimbursed through social security, accident medical expenses are compensated at 60%.", "The accident medical reimbursement rules apply to both specific areas and other areas."]

Figure 15: The prompt for ground truth content splitting

Figure 15 and Figure 16 shows the prompt for ground truth content splitting and matching for calculating completeness.

## 任务描述
你是金融领域的文本判断专家，请考虑给你的模型回答、陈述内容，然后判断这些陈述是否可以从模型回答得到。按照给定的格式，为每个陈述提供最终结论。
严格按照以下json格式输出。输出格式如下：
```
[
    {
        "result":"是/否",
        "name":"陈述内容"
    }
]
```

## 输入
模型回答:
{text}

陈述内容:
{text}

## Task Description
You are a text judgment expert in the financial field. Please first consider the model answers and statements provided to you, and then determine whether these statements can be derived from the model answers. Provide a final conclusion for each statement according to the given format. Strictly output in the following json format. The output format is as follows:
```
[
{
"result":"Yes/No",
"name":"Statement content"
}
]
```

**Input**
Model response:
{text}

Statement content:
{text}

Figure 16: The prompt for ground truth matching.

