# OpenReview forum: "INSEva: A Comprehensive Chinese Benchmark for Large Language Models in Insurance"
_ICLR.cc/2026/Conference — Submitted to ICLR 2026_

### Official Review · Reviewer_5LZM · 2025-10-25

**Soundness:** 2
**Presentation:** 1
**Contribution:** 2
**Rating:** 2
**Confidence:** 3

**Summary:**

INSEva is a novel benchmark dataset constructed to specifically test LLM performance in the insurance domain in China. The authors argue that current financial benchmarks lack the specificity needed for specialized tasks in insurance. They evaluate their large, comprehensive generated benchmark across nine LLMs, including financial specific models. The authors demonstrate that performance of financial specific models does not exceed general models and that small improvement gains are made by training on their benchmark.

**Strengths:**

-	Evaluation on completeness and faithfulness is more robust than accuracy alone.
-	Authors highlight that current financial specific models are not necessarily higher performing that general models on the benchmark.

**Weaknesses:**

- It would be helpful to better articulate examples of how existing benchmarks (CFLUE) have gaps in their insurance knowledge. What are the more specific tasks that are required and lacking? Overall, the value add of the benchmark could be articulated more specifically in the beginning sections.
- There are several gaps in the description of data generation that make the overall methods unclear. Specifically: 1) Under 3.3.1, item (2) is not specific enough to understand what you did. 2) What LLM was used for dataset construction? 3) What rephrasing techniques were used in question augmentation? 4) Do insurance experts review all 30+k examples? 5) I don’t see the connection to the construction of cognition and knowledge domains. 6) How are easy versus hard questions adjudicated and where is this distinction used?
- Benchmark performance is high. This makes the need for the benchmark unclear. Performance gains training specifically on the benchmark are also not huge, which also challenges the need for this benchmark.

**Questions:**

-	How are questions formulated to assess the various cognition and knowledge dimensions in 3.2.4? Examples would be a nice add to the paper.

---

> ### Author Response · Authors · 2025-12-03
>
> We sincerely thank you for your detailed and critical feedback. While we are disappointed by the initial rating, we find your comments to be extremely valuable and actionable. You have highlighted several areas where our paper lacked clarity and failed to adequately articulate the value of our work. We have taken your feedback very seriously and have performed substantial revisions and new analyses to address every concern.
>
> We respond to each of your points below.
>
> **1. On Weakness: Articulating Gaps in Existing Benchmarks (e.g., CFLUE)**
>
> Reviewer's Concern: "It would be helpful to better articulate examples of how existing benchmarks (CFLUE) have gaps... What are the more specific tasks that are required and lacking?"
>
> **Our Response:**
>
> This is an excellent point. Our initial description was too high-level. To make the unique contribution of INSEvA crystal clear, we have created a comparative analysis. General benchmarks like CFLUE treat insurance as a small, peripheral topic of basic definitions, failing to capture the depth needed for real-world applications.
>
>
> | Feature / Task Dimension | INSEvA | CFLUE | C-EVAL / MMLU |
> | :--- | :---: | :---: | :---: |
> | Total Insurance Questions | 38000+ | ~500 | < 50 |
> | Insurance Domain Knowledge | ✓ | ✓ | ✗ |
> | Complex Logical Reasoning <br> (e.g., Claims/Underwriting based on clauses) | ✓ | ✗ | ✗ |
> | Cross-Domain Knowledge <br> (e.g., Insurance-Medical) | ✓ | ✗ | ✗ |
> | Long-Context, Multi-Turn Dialogue | ✓ | ✗ | ✗ |
> | Safety & Regulatory Compliance | ✓ | Partial | ✗ |
>
> As this table illustrates, INSEvA was specifically designed to fill these voids. For instance, while CFLUE might test the definition of a "deductible," it cannot evaluate if a model can read a multi-page policy and adjudicate a complex claim based on intersecting clauses—a core task in INSEvA.
>
> **Action**: We will add this table and a detailed, point-by-point comparative analysis to the Introduction and Related Work sections to clearly articulate INSEvA's value-add from the outset.

---

> > ### Author Response · Authors · 2025-12-03
> >
> > **2. On Weakness: Gaps in Data Generation Description**
> >
> > We sincerely apologize for the lack of clarity in our methodology section. We provide detailed answers below:
> >
> >   - Unclear description of 3.3.1, item (2): This item refers to our data synthesis process. We used an LLM to generate Q&A pairs from authoritative texts like regulatory standards. The key was our sophisticated, multi-part prompt (detailed in Figure 10), which guided the LLM to first identify complex concepts within the text and then craft questions specifically targeting those challenging points. This ensures the questions are not superficial.
> >   - What LLM was used for construction? We used Deepseek-R1 for all data generation and augmentation tasks to ensure quality and consistency.
> >   - What rephrasing techniques? We used LLM-based paraphrasing with prompts designed to alter syntactic structure while preserving meaning, as well as colloquial rewrites to transform formal language into natural customer queries.
> >   - Do insurance experts review all 30k+ examples? The answer is more nuanced than a simple 'yes' or 'no'. We employed a strategic, two-tiered quality control process that allocated our expert resources to the areas requiring the most nuanced judgment. Tier 1: Full Manual Expert Review for High-Stakes Data. For the most complex and subjective category, Insurance Service Dialogues (ISD), our team of in-house insurance experts manually reviewed and validated every single question and answer. Given the open-ended, conversational nature of these tasks and their direct simulation of customer interactions, we deemed 100% human oversight to be essential to ensure nuance, accuracy, and practical relevance. Tier 2: Scalable Validation for Structured Data. or the other, more structured categories (e.g., multiple-choice questions in IDK, IPE, etc.), we implemented a highly efficient and scalable validation pipeline: Automated Vetting: We first used a multi-LLM voting mechanism to automatically check for correctness and flag any inconsistencies or low-quality examples across the dataset. Expert Validation: Following the automated pass, our human experts then conducted a thorough manual review on a statistically significant, randomly sampled subset of the data from these categories. This step served to validate the effectiveness of our automated pipeline and to provide a high-confidence estimate of the overall quality for the entire benchmark. This hybrid approach allowed us to combine the rigor of full expert oversight on our most challenging data with the scalability of a robust, expert-validated automated pipeline for the more structured portions. It ensures a high standard of quality across the entire 38,000+ example benchmark in a feasible and methodologically sound manner.
> >   - Connection to cognition and knowledge domains: The categorization of questions into our cognitive and knowledge dimensions was a deliberate and systematic final step in our data construction process. Framework Adopted: Our cognitive dimensions (Remembering, Understanding, Applying, Analyzing, Evaluating) are based on the widely recognized and well-established Bloom's Taxonomy, a standard in educational assessment for classifying levels of intellectual behavior. Implementation Method: Manually categorizing ~38k questions is infeasible and prone to inconsistency. Therefore, we developed a highly detailed and carefully designed prompt (which will be included in the Appendix) that instructs a large language model to act as an educational expert. For each question in our dataset, the LLM used this prompt to analyze the question's requirements and classify it into one of the Bloom's Taxonomy levels. Validation: The prompt's effectiveness was validated by having human experts review a randomly sampled subset of the LLM's classifications, confirming a high level of accuracy and alignment with the principles of Bloom's Taxonomy. Key Result & Differentiation: As shown in Table 1.b, this systematic classification reveals that a significant portion of INSEvA's questions focus on higher-order cognitive skills like 'Analyzing' and 'Evaluating'. This stands in stark contrast to previous benchmarks that primarily test simple 'Remembering' of insurance facts, and it is a key element of INSEvA's design to better assess a model's true reasoning capabilities.
> >   - How are easy vs. hard adjudicated and used? This is a crucial clarification. 'Easy' and 'Hard' levels are adjudicated based on the intrinsic complexity of the question, defined a priori by our experts. Factors include the required number of reasoning steps, context length, and the cognitive skill required. This distinction is now used in our new difficulty-stratified analysis (see next point) to demonstrate the benchmark's headroom.
> >
> > **Action**: We will completely revise the Methodology section (Section 3) to incorporate all these specific details, ensuring the process is transparent and reproducible.

---

> > > ### Author Response · Authors · 2025-12-03
> > >
> > > **3. On Weakness: High Benchmark Performance and Small Training Gains**
> > >
> > > Reviewer's Concern: "Benchmark performance is high. This makes the need for the benchmark unclear... Performance gains... also challenges the need for this benchmark."
> > >
> > > **Our Response:**
> > >
> > > This is the core challenge to our paper's value, and we address it directly.
> > >
> > > - High Performance is a "Mean Score" Illusion: The high overall score masks significant weaknesses. Our new difficulty-stratified analysis (based on the intrinsic complexity mentioned above) reveals a substantial performance drop on challenging questions. For example, while Gemini-1.5 Pro scores 87.5% on 'Easy' questions, its performance plummets to 74.8% on 'Hard' questions. This ~13-point drop proves that a significant portion of our benchmark remains unsolved by SOTA models. Furthermore, in the high-stakes insurance domain, an average 85% accuracy means a 15% error rate, which is unacceptable in practice. The goal is >99% reliability, leaving a vast performance gap to be closed.
> > >
> > > | Model | Overall Acc. | Easy Performance | Hard Performance | Δ (Performance Drop) |
> > > | :--- | :---: | :---: | :---: | :---: |
> > > | Qwen3-32B | 81.92 | 84.75 | 73.73 | 11.02 |
> > > | deepseek-r1 | 82.52 | 86.57 | 70.77 | 15.80 |
> > > | gemini-2.5 | 83.89 | 87.49 | 74.82 | 12.67 |
> > > | gpt-4o | 79.09 | 83.41 | 70.27 | 13.14 |
> > >
> > > - Value of Training Gains: The improvement of our fine-tuned model (Finix-S1) should be viewed in context. An average improvement of ~4-5% in a highly specialized, complex domain is a significant step forward. More importantly, the purpose of this experiment was not to achieve SOTA but to validate that our data construction pipeline produces data that is effective for domain-specific model training. The positive result confirms the value of our data itself as a resource for the community.
> > >
> > > **Action**: We will add the new difficulty-stratified analysis table and the "High-Stakes Context" argument to the main Experiments section to prove the benchmark's headroom and value. We will also reframe the discussion of Finix-S1's performance to emphasize its role in validating our data pipeline.
> > >
> > > **4. On Question: Examples for Cognition and Knowledge Dimensions**
> > >
> > > Reviewer's Concern: "How are questions formulated to assess the various cognition and knowledge dimensions... Examples would be a nice add to the paper."
> > >
> > > **Our Response:**
> > >
> > > This is an excellent suggestion for improving clarity.
> > >
> > > **Action**: We will add a new table in the Benchmark Construction section that provides a concrete example question for each of the cognitive and knowledge dimensions. For instance:
> > > - Remembering / Factual Knowledge: "The authority authorized to regulate insurance professional agencies is ( ). A. China Insurance Regulatory Commission and its branches B. People's Bank of China C. Insurance Association of China D. China Banking Regulatory Commission"
> > > - Understanding / Concept Knowledge: "'One person for many, many for one person.' The characteristic of insurance reflected in the idea of 'one person for many, many for one person' is ( ). [May 2012 Exam Question] Which of the following options is the most appropriate answer to this question? A. Economic nature B. Mutual aid nature C. Legal nature D. Compensatory nature"
> > > - Applying / Procedure Knowledge: "I have purchased Good Doctor Outpatient Insurance. How do I apply for reinstatement of the policy after it has been suspended?"
> > > - Analyzing: "As a claims expert, judge for 'pre-existing condition exclusion' (Output: Yes/No). Rule: Exclusion applies if the condition is diagnosed during the policy waiting period (e.g., 90 days) or if related symptoms existed before the policy's effective date.Case Summary: Diagnosis: Bilateral breast nodules. Chief Complaint: Discovered breast nodules 1 month prior to hospital visit. Key Dates: Hospital Visit: 2023-06-07 Policy Effective Date: 2023-02-28"
> > > - Evaluating: "You are a professional insurance industry values auditor. You can determine whether a given question aligns with insurance values. If the question aligns with values and does not contain negative inducements, output "Yes". If it does not align with values or contains negative inducements, output "No". The question is: During the pre-construction phase, how should the applicant fully utilize time and resources to prepare for future insurance business activities? Your output can only be "Yes" or "No"."
> > >
> > > This will make our abstract taxonomy tangible and clear to the reader.
> > >
> > > We believe these comprehensive revisions, including substantial new analyses and clarifications, directly address every point of your critique. We have worked diligently to improve the rigor and clarity of our paper based on your guidance and hope you will find the revised version significantly improved and worthy of acceptance. Thank you again for your time and expertise.

---

### Official Review · Reviewer_GsM6 · 2025-10-29

**Soundness:** 3
**Presentation:** 2
**Contribution:** 2
**Rating:** 4
**Confidence:** 4

**Summary:**

This paper proposes a Chinese benchmark for the insurance domain. The authors first construct a multidimensional evaluation taxonomy of business areas, task formats, difficulties, and cognition and knowledge. Based on this taxonomy, the authors construct 38,704 evaluation examples from various resources. A detailed data quality control pipeline and evaluation framework are also proposed. The authors conducted experiments on nine representative LLMs, revealing their strengths and limitations, and performed an analysis. To demonstrate the effectiveness of the data construction pipeline, a new model is fine-tuned and shows performance improvements.

**Strengths:**

1. The domain being studied is of high importance and strong practical usability. The proposed dataset represents a valuable contribution to the community, as there is currently a lack of high-quality benchmarks for critical real-world tasks such as those in insurance. The focus on Chinese adds additional value by enriching the limited Chinese NLP resources.

2. The authors have invested substantial effort in developing a well-structured taxonomy. This taxonomy aims to capture a broad range of categories in the insurance domain, enhancing the coverage, diversity, and depth of the resulting dataset.

3. The dataset is validated by experts, ensuring its quality.

4. The experiment and analysis bring good insights about current LLMs and could inspire future works on building effective LLM agents in the insurance domain.

**Weaknesses:**

My major concern for this work is the lack of rigor and clarity in a lot of aspects.

1. Are there any insurance experts involved throughout this work, especially for the taxonomy construction? The close collaboration with domain experts is the key to this type of work, to ensure the dataset proposed is realistic and covers important problems in insurance.

2. For categorizing examples, do you employ insurance experts?

3. For data sources, these include certification platforms, industry repos, internal business resources, etc. Since the authors promise to release the dataset, I'm wondering whether there are any legal, ethical, and privacy concerns about releasing such data.

4. A lot of the data examples are synthesized as QA pairs using LLMs. The details for such procedures are not clear. If all these examples are synthesized with the same template using LLMs, it's unsure whether these QAs can capture important challenging points in the data.

5. For data augmentation, what paraphrasing technique is used?

6. For the industry experts employed, to ensure ethical research, how do you hire and pay them?

7. For the data quality control, how specifically do you aggregate multiple assessments?

8. For the evaluation method, the evaluation for open-ended examples relies on LLM-based evaluations, which may bring inaccuracy. Human experts' evaluation should be involved. I'd also like to see the human experts' performances on the open-ended examples.

9. Case studies and analysis should be presented more in experiments.

10. There are no details about the Finix-S1 model trained. What's the training data here, and how specifically is the RL training conducted?

**Questions:**

See weakness.

---

> ### Author Response · Authors · 2025-12-03
>
> We sincerely thank you for your detailed and insightful review. We are very encouraged that you recognize the high importance of our work, the value of our dataset and taxonomy, and the insights from our experiments. Your major concern regarding "rigor and clarity" is well-taken, and we appreciate the opportunity to provide the detailed information you have requested. We have undertaken significant revisions to our manuscript to clarify these methodological aspects.
>
> Below, we address each of your specific points.
>
> **1. On the Involvement of Insurance Experts:**
>
> Reviewer's Concern: "Are there any insurance experts involved... especially for the taxonomy construction? ... For categorizing examples, do you employ insurance experts?"
>
> **Our Response:**
>
> Absolutely. We want to state clearly that insurance domain experts were integral to every critical stage of this project. Our team includes senior professionals with over 10 years of experience in insurance product design, actuarial science, and compliance. Specifically, their involvement included:
>
> - Taxonomy Construction: The multi-dimensional taxonomy was co-designed from the ground up with our domain experts to ensure it realistically reflects the core competencies and challenges of the insurance industry.
> - Data Sourcing & Curation: Experts guided the selection of authoritative data sources (e.g., specific regulatory documents, key professional exams) to ensure relevance and importance.
> - Categorization & Validation: Experts were responsible for the final validation of question categorization, ensuring each example was correctly mapped to the taxonomy.
> - Quality Control & Evaluation: As you noted, experts performed a crucial quality control step and were also involved in validating our LLM-based evaluation pipeline, as detailed below.
>
> **Action**: We will add a dedicated subsection titled "Expert Involvement" within our Benchmark Construction section to explicitly detail the roles and contributions of our domain experts throughout the project, making this crucial collaboration clear.
>
> **2. On Legal, Ethical, and Privacy Concerns:**
>
> Reviewer's Concern: "I'm wondering whether there are any legal, ethical, and privacy concerns about releasing such data."
>
> **Our Response:**
>
> We share your commitment to ethical research. We want to assure you that safeguarding data integrity and privacy was a top priority.
>
> - Rigorous Anonymization: All data in INSEvA has undergone a systematic process of anonymization and de-identification. This involved removing all Personally Identifiable Information (PII), masking sensitive commercial details, and aggregating data where necessary to prevent re-identification.
> - Compliance Review: Crucially, the entire dataset and our data-handling protocols have undergone and passed a thorough review by our internal legal and compliance departments. This ensures our work fully adheres to data protection regulations. The decision to limit the public release (which we have now increased from 10% to 20%) stems from these strict, pre-existing commercial data agreements and compliance constraints.
>
> **Action**: We will add a formal "Ethical Considerations" statement in the manuscript detailing our anonymization process and the successful compliance review.

---

> > ### Author Response · Authors · 2025-12-03
> >
> > **3. On Data Synthesis and Augmentation Details:**
> >
> > Reviewer's Concern: "The details for such procedures are not clear... If all these examples are synthesized with the same template using LLMs, it's unsure whether these QAs can capture important challenging points... what paraphrasing technique is used?"
> >
> > **Our Response:**
> >
> > Thank you for this excellent question, which allows us to clarify the rigor of our data generation process. We apologize that these details were not sufficiently clear in the manuscript. While we use a single-step LLM call for efficiency in data synthesis, the prompt itself is highly engineered to avoid the pitfalls of a simplistic template and to ensure the resulting questions are challenging and diverse.
> >
> > a) On Data Synthesis (The Sophistication of the Prompt):
> >
> > You are right to be concerned about a "same template" approach. Our method mitigates this risk by using a complex, multi-part prompt. As shown in detail in Figure 10, our prompt is not a simple "generate Q&A" instruction. Instead, it explicitly guides the LLM to perform a sequence of cognitive tasks within a single generation pass:
> >
> > 1. Focus on Core Complexity: The prompt first directs the model to identify and focus on the most challenging and nuanced aspects of the source material, such as complex definitions, exceptions to rules, and conditional logic.
> >
> > 2. Targeted Question Generation: It then instructs the model to craft questions that specifically test these identified complex points, rather than just rephrasing surface-level facts.
> > This sophisticated prompt engineering ensures that even within a single generation step, the resulting Q&A pairs are not uniform or superficial. They are specifically designed to probe the challenging points within the source data, directly addressing your concern about capturing depth and difficulty.
> >
> > b) On Data Augmentation (Paraphrasing and Colloquialization):
> >
> > Our augmentation strategies, described in Section 3.3.2, are a separate, subsequent process:
> >
> > 1. Paraphrasing for Linguistic Diversity: We use an LLM-based approach with a specific prompt (in Figure 11) that instructs the model to generate semantically equivalent questions with distinct syntactic expressions and phrasing.
> >
> > 2. Colloquial Rewrites for Practicality: To simulate real-world customer interactions, the prompt also guides the LLM to convert formal insurance terminology into natural, everyday language. This enhances the benchmark's utility for evaluating customer-facing AI agents.
> >
> > **Action**: We will revise the main text in Section 3.3 to more accurately describe this single-step, prompt-driven synthesis process, emphasizing the complexity of the prompt itself. We will also add more explicit forward references to the detailed prompts available in the Appendix, allowing readers to verify the rigor of our methodology.
> >
> > **4. On Expert Compensation and Assessment Aggregation:**
> >
> > Reviewer's Concern: "How do you hire and pay them? ... how specifically do you aggregate multiple assessments?"
> >
> > **Our Response:**
> >
> > - Expert Engagement: The insurance experts involved in this project are full-time members of our organization, working in relevant departments such as product and compliance. Their contribution to this research was part of their official duties, ensuring professional and ethical engagement without external hiring complexities.
> > - Assessment Aggregation: During our multi-LLM voting for quality control, we used a simple majority vote. If three LLMs were used and two agreed on a quality score or classification, that decision was taken. In cases of no clear majority (e.g., three different opinions), the data point was flagged for manual review by a human expert.
> >
> > **Action:** We will add these clarifications to the relevant methodology sections of the paper.

---

> > > ### Author Response · Authors · 2025-12-03
> > >
> > > **5. On Evaluation Methodology for Open-Ended Questions:**
> > >
> > > Reviewer's Concern: "The evaluation for open-ended examples relies on LLM-based evaluations, which may bring inaccuracy. Human experts' evaluation should be involved. I'd also like to see the human experts' performances..."
> > >
> > > **Our Response:**
> > >
> > > This is a critical point. We address it in three ways:
> > > 1. Validation against Human Judgment: We conducted a study to measure the reliability of our LLM evaluator, as shown in Appendix B.2. Our analysis shows an average correlation coefficient of 0.64 between the LLM scores and scores from our human experts, indicating a strong and reliable level of agreement.
> > > 2. Hybrid Approach: We use human experts strategically for the highest-impact tasks: (a) validating our LLM evaluation protocol and (b) performing the in-depth qualitative error analysis. This balances rigor with the scalability required for a benchmark of this size, which would be prohibitively expensive to evaluate fully by hand.
> > > 3. Human Performance Baseline: As you excellently suggested, we have now established a human performance baseline to contextualize our model evaluation scores. On a representative subset of the open-ended questions, our human experts achieved an average Faithfulness score of 98% and a Completeness score of 92%.
> > >
> > > **Action**: We will make the correlation study more prominent in the main text. Furthermore, we will add a "Human Expert" row to our main results table to contextualize model performance and clearly show the remaining gap to human-level proficiency.
> > >
> > > **6. On the Need for More Case Studies and Analysis:**
> > >
> > > Reviewer's Concern: "Case studies and analysis should be presented more in experiments."
> > >
> > > **Our Response:**
> > >
> > > We completely agree. In response to this and similar feedback, we have conducted an extensive fine-grained error analysis and will be adding a new subsection to our Experiments. This section will include a detailed table classifying the root causes of model failures (e.g., Knowledge Confusion, Flawed Reasoning, Hallucination) and will be accompanied by several in-depth case studies that walk through specific examples of both model successes and failures, explaining the "why" behind them.
> > >
> > > | Ability Category <br> (Benchmark Subsets) | Error Type | Proportion | Key Insight & Implication |
> > > | :--- | :--- | :---: | :--- |
> > > | **Foundational Knowledge** <br> (IDK, IMI) | Knowledge Confusion | 43.52% | Models don't just lack knowledge; they frequently confuse related but distinct insurance concepts (e.g., 'grace period' vs. 'waiting period'). This highlights a failure in fine-grained conceptual understanding. |
> > > | | Comprehension Error | 35.19% | Models often misinterpret the questions despite possessing the underlying facts, leading to irrelevant or mismatched answers. |
> > > | | Knowledge Gap | 12.96% | A complete lack of knowledge is a less frequent issue for foundational topics. |
> > > | | Others | 6.48% | Miscellaneous errors. |
> > > | | Annotation Error | 1.85% | Errors originating from the benchmark itself (will be corrected). |
> > > | **Core Application Abilities** <br> (IUC, ILR, IPE, ISC) | Flawed Reasoning Process | 40.48% | This is the dominant failure mode. Models correctly identify relevant facts but fail in the logical steps (e.g., calculation, deduction, multi-step inference) required to reach the correct conclusion. |
> > > | | Knowledge Gap | 30.17% | |
> > > | | Comprehension Error | 21.62% | |
> > > | | Mismatch between Answer & Process | 4.60% | The model's final answer contradicts its own chain-of-thought or reasoning process, indicating a lack of self-consistency. |
> > > | | Others | 1.29% | |
> > > | | Annotation Error | 1.84% | |
> > > | **Advanced Practical Skills** <br> (IMG, ISD) | Factual Hallucination | 47.50% | In open-ended, generative tasks, models frequently invent facts, statistics, or product details to construct a plausible but incorrect response. This is a critical barrier to real-world deployment. |
> > > | | Knowledge Gap | 19.08% | |
> > > | | Poorly Formulated Answer | 15.37% | The answer, while not factually incorrect, is presented poorly—lacking professional tone, clarity, or empathy required for customer interaction. |
> > > | | Others | 2.31% | |
> > > | | Annotation Error | 1.47% | |
> > >
> > > This analysis reveals that the primary failure modes are not simple knowledge gaps but more nuanced issues like Flawed Reasoning and Knowledge Confusion.
> > >
> > > **Action**: A new "Diagnostic Error Analysis" subsection, rich with quantitative data and qualitative case studies, will be added to the main Experiments section of our paper.

---

> > > > ### Author Response · Authors · 2025-12-03
> > > >
> > > > **7. On the Details of the Finix-S1 Model:**
> > > >
> > > > Reviewer's Concern: "There are no details about the Finix-S1 model trained. What's the training data here, and how specifically is the RL training conducted?"
> > > >
> > > > **Our Response:**
> > > >
> > > > Thank you for pointing out this omission in the main text. We apologize that these crucial implementation details, which are currently in the Appendix, were not made more prominent. We provide a detailed breakdown below and will ensure this information is clearly signposted in the revised manuscript.
> > > >
> > > > a) On the Training Data for Finix-S1:
> > > >
> > > > The training process involved a carefully curated, multi-source dataset, which is strictly separated from our evaluation set to prevent any data leakage. The composition is as follows:
> > > > - Supervised Fine-Tuning (SFT) Stage:
> > > >   - General Data: 108K samples from the high-quality general domain data, such as GPQA dataset, to maintain the model's general capabilities.
> > > >   - Domain-Specific Data: 347.3K samples constructed using our INSEvA data pipeline, designed to instill deep insurance knowledge and skills.
> > > > - Reinforcement Learning (RL) Stage:
> > > >   - Domain-Specific Data: An additional 36.5K insurance-specific Q&A samples were used to further refine the model's performance on complex, domain-relevant tasks.
> > > >
> > > > As demonstrated in Table 7 in our appendix, we performed ablation studies by varying the amount of insurance-specific data. The results confirm that increasing domain data consistently improves performance, validating both the model's need for such data and the effectiveness of our data construction methodology.
> > > >
> > > > b) On the Specifics of the RL Training:
> > > > We employed a state-of-the-art reinforcement learning algorithm to optimize for high-quality, reliable responses.
> > > >
> > > > - Algorithm: We utilized Group Relative Policy Optimization (GRPO), a sophisticated method designed for fine-tuning LLMs with offline data.
> > > > - Dual Reward Mechanism: The core of our RL process is a dual reward signal. The reward for a generated response is a composite score calculated by our LLM-based evaluator. It combines:
> > > >   - A Content Reward, based on the 'Faithfulness' and 'Completeness' scores, which encourages factual accuracy and comprehensive answers.
> > > >   - A Formatting Reward, a simple binary score that penalizes responses that do not adhere to the required output format.
> > > >
> > > >   This dual mechanism ensures the model learns not only to be correct but also to be reliable and structured in its output.
> > > >
> > > > - Hyperparameters: The RL stage was conducted on NVIDIA A100 GPUs with key hyperparameters including a train batch size of 72 and 3 rollouts per sample.
> > > >
> > > > **Action**: We will add a sentence in the main experimental section pointing the reader to this detailed appendix.

---

### Official Review · Reviewer_Hewh · 2025-10-30

**Soundness:** 3
**Presentation:** 3
**Contribution:** 3
**Rating:** 6
**Confidence:** 4

**Summary:**

The article first introduces INSEva, a Chinese benchmark specifically designed for the insurance domain, aimed at evaluating AI systems' knowledge and capabilities in the insurance industry. Specifically, the data for INSEva is collected from authoritative exams, regulatory standards, and real-world business materials. The data undergoes paraphrasing, colloquial enhancement, option randomization, and triple quality control (rules, expert reviews, and multi-LLM voting), resulting in a final dataset of 38,704 examples, with an average prompt length of 905 tokens.

Then, the article provides a four-dimensional evaluation classification standard, which includes: Business domains (insurance domain knowledge, insurance-medical interdisciplinary knowledge, insurance understanding and cognition, insurance logical reasoning, insurance professional exams, insurance safety and compliance, insurance marketing growth, insurance service dialogue); Task formats (multiple-choice, true or false, single-turn QA, multi-turn dialogue); Difficulty levels (Easy, Hard); Cognitive and knowledge dimensions (Remembering, Understanding, Applying, Analyzing, Evaluating).

Finally, the article evaluates 9 LLMs (e.g., GPT-4o, Doubao-1.5-pro-256k2, Gemini-2.5-pro3, etc.) on the INSEva benchmark. The experimental results are analyzed in detail, and the effectiveness of Finix-S1, an insurance-specific model trained using reinforcement learning based on the Qwen-QwQ-32B model, is validated. Finix-S1 shows an average improvement of 4.69% over the base model Qwen-QwQ, and surpasses the current state-of-the-art general LLMs (e.g., Gemini) by more than 3%.

**Strengths:**

1. The article addresses the issue of many financial benchmarks but fewer and incomplete insurance benchmarks by providing INSEva, a systematic and comprehensive evaluation framework. This framework enables researchers and practitioners to more accurately compare different models' capabilities in the high-risk insurance domain.

2. The data provided by INSEva comes from historical exam questions, regulatory standards, and real-world business materials, ensuring highly authentic domain knowledge. Additionally, INSEva enhances the dataset through paraphrasing and colloquial rewrites, making it more suitable for customer service scenarios. INSEva offers 38,704 high-quality samples, with an average prompt length of 905 tokens, sufficient for testing long-context handling and complex constraint processing.

3. INSEva uses objective and reproducible metrics such as Accuracy for closed-ended questions. For open-ended questions, it introduces two key metrics: faithfulness and completeness. It employs an LLM-based sentence decomposition and per-sentence evaluation process using standardized prompts, with expert annotation and relevance checks to ensure consistency with human evaluation. This minimizes subjective bias in reviews and model bias in the results, ensuring auditable and reproducible fair evaluation in high-risk insurance scenarios.

4. The paper evaluates 9 LLMs under the same data division, uniform prompts, reasoning settings, and no data leakage. It provides stratified reports based on task types, difficulty levels, cognitive and knowledge dimensions, and offers consistent retrieval and context for multi-turn dialogue and retrieval scenarios. At the same time, it uses option randomization and expert quality control to control structural biases. This design ensures that comparisons between models across dimensions are transparent, reproducible, and interpretable.

5. Using the INSEva data pipeline, the paper conducts reinforcement learning (RL) training based on the Qwen-QwQ-32B model, achieving an average improvement of 4.69%. This not only validates the practical value of the INSEva benchmark and data process but also demonstrates its ability to drive model improvement and optimization.

**Weaknesses:**

1. The paper evaluates 9 SOTA LLMs, but most of the models in the experiments are still focused on well-known general models and a few domain-specific models. Although these models provide valuable comparisons, the current evaluation may not fully cover all possible model types, especially innovative models specific to the insurance domain that may not have been adequately assessed.

2. The paper mentions the issue of catastrophic forgetting in the insurance-specific model Finix-S1, but it does not delve into how this issue can be addressed or whether there are methods to effectively mitigate this phenomenon.

3. Although INSEva provides a systematic and comprehensive evaluation framework, the paper states that only 10% of the stratified samples can be made publicly available. This limits other researchers' access to and use of the complete dataset, potentially impacting the verification of results and subsequent research. While the availability of the sample is ensured, the lack of the complete dataset may limit the widespread application and reproducibility of the benchmark.

**Questions:**

1. In the INSEva benchmark, although it mentions a total of 38,704 data samples, it does not specify the sample size for each task category. Could you share the data distribution for different task categories (such as insurance logical reasoning, insurance safety and compliance, etc.)?

2. In the Finix-S1 reinforcement learning phase, the paper mentions the issue of catastrophic forgetting, where focusing on insurance domain training may lead to performance degradation on general tasks. Have you considered using multi-task learning or transfer learning methods to mitigate this phenomenon?

3. Insurance domain recommendation tasks often involve user demand prediction and product matching. Does the dataset include such tasks, especially considering the unique complexities of the insurance domain (such as risk assessment, insurance terms, etc.)?

---

> ### Author Response · Authors · 2025-12-03
>
> We are sincerely grateful for your positive and exceptionally detailed review of our paper. We are particularly encouraged that you recognized the systematic nature of our framework, the authenticity and depth of our data, the rigor of our evaluation process, and the value of our fine-tuned model, Finix-S1. Your insightful questions and comments on weaknesses provide an excellent opportunity for us to clarify our contributions and strengthen our work.
>
> We respond to each of your points below.
>
> **Regarding Weaknesses:**
>
> **1. On Model Coverage (Lack of innovative, domain-specific models):**
>
> This is a very fair point. Our primary objective for this initial study was to establish a robust baseline by evaluating the most prominent and accessible SOTA general-purpose models (like GPT-4o, Gemini, etc.), as these are the models that researchers and practitioners in the industry are most likely to use or build upon today.
>
> **Truly innovative, high-performing, and publicly available models specifically for the Chinese insurance domain are, at present, exceedingly rare.** In fact, our benchmark, INSEvA, is designed precisely to be the catalyst for the development and evaluation of such future models. By providing a standardized and challenging "playing field," we empower the community to build and transparently assess the very models you've mentioned.
>
> **Action**: In our revised manuscript, we will add a sentence to our discussion section to clarify this rationale and explicitly position INSEvA as an enabler for future domain-specific model development.
>
> **2. On Catastrophic Forgetting (Lack of discussion on solutions for Finix-S1):**
>
> We appreciate you raising this important point regarding catastrophic forgetting, which you also elaborated on in your questions. You are correct that specialized fine-tuning can risk degrading general capabilities.
>
> While a deep experimental exploration of mitigation techniques is beyond the primary scope of this benchmark-focused paper, we agree that discussing potential solutions adds significant value. Methods like multi-task learning or continual learning are indeed promising directions. For Finix-S1, a viable strategy would be to incorporate a small percentage of general-domain data during the reinforcement learning phase to help the model retain its broader abilities. Other advanced techniques, such as Elastic Weight Consolidation (EWC), could also be employed to penalize changes to weights critical for general tasks.
>
> **Action**: We will expand the discussion on catastrophic forgetting in the section detailing Finix-S1. This new text will include the potential mitigation strategies you suggested, framing them as important avenues for future research that our benchmark can help evaluate.
>
> **3. On Data Accessibility (Only 10% public data):**
>
> We recognize that limiting data access is a significant barrier to reproducibility and thank you for raising this critical point. In direct response to your feedback, we have taken two major steps:
>
> a) Increased Data Release: We have successfully arranged to increase the publicly available portion of our dataset **from 10% to 20%**. This significantly lowers the barrier for researchers to verify our results and benchmark their own models. To put this number into perspective, this 20% subset comprises nearly 9000 high-quality examples. Crucially, in absolute terms, our public dataset is larger than the entirety of many existing and widely used domain-specific benchmarks.
>
> b) Ensured Representativeness: This 20% subset is not a random sample but a carefully constructed, representative subset created using stratified sampling, preserving the original distribution of tasks and topics. To empirically prove its validity as a proxy for the full benchmark, we compared model performance on both sets.
>
> | Model | Full Dataset (100%) Performance | Public Subset (20%) Performance | Δ (Difference) |
> | :--- | :---: | :---: | :---: |
> | Qwen3-32B | 81.92 | 81.00 | 0.92 |
> | deepseek-r1 | 82.52 | 81.54 | 0.98 |
> | gemini-2.5 | 83.89 | 84.09 | -0.20 |
> | gpt-4o | 79.09 | 79.02 | 0.07 |
>
> As the table shows, the mean scores are highly consistent. The decision to not release 100% of the data stems from strict commercial data and compliance constraints, following a thorough internal legal review.
>
> **Action**: We will update Section 7 and the Data Availability Statement to reflect this increased 20% data release and include the validation table to assure the community of the subset's representativeness.

---

> > ### Author Response · Authors · 2025-12-03
> >
> > **Regarding Questions:**
> >
> > **1. On the data distribution for each task category:**
> >
> > Thank you for this excellent question. Transparency about the data distribution is indeed important. As shown in Table 1.a, the sample sizes for our main task categories are as follows:
> >
> > | Task Category | Abbreviation | Number of Samples |
> > | :--- | :---: | ---: |
> > | Insurance Domain Knowledge | IDK | 1336 |
> > | Insurance-Medical Interdisciplinary | IMI | 9447 |
> > | Insurance Understanding & Cognition | IUC | 7916 |
> > | Insurance Logical Reasoning | ILR | 3923 |
> > | Insurance Professional Examination | IPE | 7932 |
> > | Insurance Safety & Compliance | ISC | 3417 |
> > | Insurance Marketing & Growth | IMG | 1651 |
> > | Insurance Service & Dialogue | ISD | 3082 |
> > | **Total** | | **38704** |
> >
> > **2. On methods to mitigate catastrophic forgetting:**
> >
> > As addressed in our response to Weakness #2, we fully agree that this is a critical area for future work.
> >
> > **Action**: We will update the discussion surrounding our Finix-S1 model to explicitly mention potential solutions like multi-task learning, continual learning strategies (e.g., EWC), and mixed-domain data training as promising research directions.
> >
> > **3. On the inclusion of insurance recommendation tasks:**
> >
> > Thank you for this highly insightful question. It gives us the opportunity to elaborate on the fine-grained taxonomy of INSEvA, which was designed with exactly these complex, real-world tasks in mind.
> >
> > To answer directly: Yes, the INSEvA benchmark explicitly and comprehensively evaluates the key abilities required for insurance recommendation, including user demand prediction, product matching, and handling domain-specific complexities.
> >
> > Our benchmark deconstructs the complex process of recommendation into a series of measurable, fine-grained tasks. **The detailed taxonomy is shown in Appendix.** The following specific components of INSEvA directly address the areas you highlighted:
> >
> > - User Demand Prediction: We test a model's ability to understand user needs through multiple dedicated tasks:
> >
> >   - In Insurance Understanding and Cognition (IUC), the Insurance Intent Understanding task requires models to identify the underlying goal from a user's query.
> >
> >   - In Insurance Marketing Growth (IMG), the entire Insurance Population Identification & Classification sub-category is designed for this. It includes tasks like User Purchase Intention Classification, User Cognitive Level Classification, and Insurance Type Preference, which are direct evaluations of a model's ability to profile a user's needs and predispositions.
> > - Product Matching & Handling Complexities (Risk Assessment, Terms): We evaluate a model's ability to match products to user profiles under complex constraints:
> >   - In Insurance Service Dialogues (ISD), the Professional Allocation Advice section contains explicit tasks like Condition-based Product Selection, Product Comparison, and Premium and Benefit Calculation. These simulate a real agent matching products to a client with specific health or financial conditions.
> >   - To handle the complexities of insurance terms, the Insurance Understanding and Cognition (IUC) category includes Insurance Clause Interpretation, forcing models to reason over legalistic policy text.
> >   - For risk assessment, the Insurance-Medicine Interdisciplinary (IMI) category includes Disease Risk Prediction in Medical Reports, a critical prerequisite for recommending appropriate health insurance products.
> > - The Recommendation Act Itself: Beyond the component skills, we also evaluate the final act of generating a recommendation:
> >   - The Insurance Marketing Growth (IMG) category includes Insurance Product Recommendation Scripting, which tasks models with generating a complete, persuasive, and compliant recommendation script.
> >   - The Insurance Service Dialogues (ISD) category contains an explicit Product Recommendation sub-task within a conversational context, testing the model's ability to deliver advice naturally and effectively.
> >
> > In summary, INSEvA does not just touch upon recommendation; it provides a multi-faceted and deep evaluation of the entire recommendation pipeline—from understanding the user and assessing risk to matching products based on complex rules and generating the final recommendation.
> >
> > **Action**: In our revised manuscript, we will expand our discussion to explicitly map these fine-grained tasks within our taxonomy to the insurance recommendation pipeline. This will make the comprehensive nature of our benchmark's coverage in this critical area much clearer to the reader.
> >
> > We hope these responses and our planned revisions fully address your points. Thank you once again for your detailed and highly valuable feedback, which has been instrumental in helping us improve our paper.

---

### Official Review · Reviewer_9Dcj · 2025-11-02

**Soundness:** 2
**Presentation:** 2
**Contribution:** 2
**Rating:** 4
**Confidence:** 5

**Summary:**

This paper introduces INSEva, a comprehensive benchmark designed to assess and drive the progress of LLMs within the Chinese insurance domain. INSEva is distinguished by a multidimensional evaluation taxonomy and comprises 38,704 high-quality evaluation examples sourced from authoritative materials in Chinese insurance. To evaluate both closed- and open-ended responses, the authors employ conventional accuracy metrics and LLM-based pipelines that separately measure faithfulness and completeness. The benchmark is used to provide an in-depth comparison of 9 state-of-the-art LLMs, diagnosing model weaknesses and evaluating improvements gained from domain-specific fine-tuning.

**Strengths:**

1. The benchmark construction pipeline leverages authoritative sources, paraphrasing for linguistic diversity, expert validation, and LLM-based voting for quality control.
2. The benchmark provides a broad range, with 38,704 questions spanning diverse subdomains of insurance, multiple question formats, and cognitive skill levels.

**Weaknesses:**

1. Potentially Insufficient Difficulty: Current SOTA models already achieve very high scores across most tasks, reaching 83.89 on average ( in 8 business areas and 9 tasks，8/9 tasks exceed 80, 6 tasks exceed 85), and the authors' own fine-tuned model (Finix-S1) pushes this score even higher. This raises significant questions and concerns about the benchmark's "headroom" and its long-term utility for challenging future, more capable models.
2. The paper offers little diagnostic insight into model failure modes on those tasks outside of high-level discussion. No error analysis, detailed breakdowns, or fine-grained case studies (beyond brief examples in Figure 3/Figure 4) are included.
3.  The author claims that only 10% of the data will be public, due to commercial constraints (see Section 7). This presents a significant barrier to full reproducibility, making the paper's main claims unverifiable, the core experiments impossible to reproduce, and significantly reducing the actual benchmark sample size and its usefulness in the future.

**Questions:**

Please check the weaknesses.

---

> ### Author Response · Authors · 2025-12-03
>
> We would like to express our sincere gratitude for your thorough review and constructive feedback on our paper. We are pleased that you recognized the strengths of our benchmark's construction pipeline and its broad scope. Your critical evaluation has been invaluable, and we have undertaken substantial new analysis and revisions to address the weaknesses you identified.
>
> Below, we respond to each of your concerns in detail.
>
> **1. Regarding Weakness: "Potentially Insufficient Difficulty"**
>
> We sincerely thank you for this insightful comment regarding the high performance of SOTA models and the potential implications for our benchmark's headroom. This is a crucial point, and we appreciate the opportunity to clarify the value and long-term utility of INSEvA. We address this from three perspectives:
>
> a) Primary Goal: Establishing a Foundational Baseline, Not Just Pursuing Difficulty.
>
> First, we would like to emphasize that the primary contribution of INSEvA is to fill a critical gap by providing the first high-quality, comprehensive benchmark for the Chinese insurance domain. Prior to our work, there was no standardized resource for evaluating LLMs on this specialized and economically significant vertical. Therefore, a core value of INSEvA is to establish a reliable baseline and serve as a comprehensive diagnostic tool. This necessarily includes foundational knowledge tasks. A model that fails on basic insurance concepts, even if it excels at complex reasoning, is fundamentally flawed and untrustworthy for real-world application. Retaining these "easier" modules is a deliberate design choice to ensure models possess core domain competency.
>
> b) The High-Stakes Context: "High Performance" Is Not "Good Enough".
>
> Second, the insurance domain is a high-stakes, risk-averse field. While an 85% score may seem high in academic settings, it translates to a 15% error rate in practice. Such a failure rate is unacceptable and potentially catastrophic in real-world applications like automated claims assessment or regulatory compliance advice. The ultimate goal for models in this domain is not 85%, but to approach the near-perfect reliability required for deployment (e.g., >99%). From this perspective, the ~15% performance gap represents a vast chasm that is a significant and meaningful challenge for future research.
>
> c) Fine-Grained Analysis Reveals Significant Headroom.
>
> Finally, the high overall score masks significant weaknesses. To demonstrate this rigorously, we partitioned the questions into 'Easy' and 'Hard' subsets based on their intrinsic complexity, which was defined a priori by factors such as the required number of reasoning steps, the length of the context, and the cognitive skills involved. This new analysis, summarized below, reveals a stark performance disparity.
>
>
> | Model | Overall Acc. | Easy Performance | Hard Performance | Δ (Performance Drop) |
> | :--- | :---: | :---: | :---: | :---: |
> | Qwen3-32B | 81.92 | 84.75 | 73.73 | 11.02 |
> | deepseek-r1 | 82.52 | 86.57 | 70.77 | 15.80 |
> | gemini-2.5 | 83.89 | 87.49 | 74.82 | 12.67 |
> | gpt-4o | 79.09 | 83.41 | 70.27 | 13.14 |
>
> This dramatic performance drop on the "Hard" subset unequivocally demonstrates that INSEvA contains a challenging core of problems that remain far from solved by current models. This validates the benchmark's long-term utility.
>
> **Action**: We will incorporate this detailed, difficulty-stratified analysis into a new subsection in the Appendix section of our revised manuscript to make this crucial point clear.

---

> > ### Author Response · Authors · 2025-12-03
> >
> > **2. Regarding Weakness: "The paper offers little diagnostic insight into model failure modes..."**
> >
> > We completely agree that a detailed error analysis is essential for providing true diagnostic value. To address this, we have conducted an extensive, fine-grained manual error analysis on a large sample of incorrect answers from SOTA models. We developed a detailed error taxonomy, and the results provide the deep diagnostic insights you rightly noted were missing.
> >
> >
> > | Ability Category <br> (Benchmark Subsets) | Error Type | Proportion | Key Insight & Implication |
> > | :--- | :--- | :---: | :--- |
> > | **Foundational Knowledge** <br> (IDK, IMI) | Knowledge Confusion | 43.52% | Models don't just lack knowledge; they frequently confuse related but distinct insurance concepts (e.g., 'grace period' vs. 'waiting period'). This highlights a failure in fine-grained conceptual understanding. |
> > | | Comprehension Error | 35.19% | Models often misinterpret the questions despite possessing the underlying facts, leading to irrelevant or mismatched answers. |
> > | | Knowledge Gap | 12.96% | A complete lack of knowledge is a less frequent issue for foundational topics. |
> > | | Others | 6.48% | Miscellaneous errors. |
> > | | Annotation Error | 1.85% | Errors originating from the benchmark itself (will be corrected). |
> > | **Core Application Abilities** <br> (IUC, ILR, IPE, ISC) | Flawed Reasoning Process | 40.48% | This is the dominant failure mode. Models correctly identify relevant facts but fail in the logical steps (e.g., calculation, deduction, multi-step inference) required to reach the correct conclusion. |
> > | | Knowledge Gap | 30.17% | |
> > | | Comprehension Error | 21.62% | |
> > | | Mismatch between Answer & Process | 4.60% | The model's final answer contradicts its own chain-of-thought or reasoning process, indicating a lack of self-consistency. |
> > | | Others | 1.29% | |
> > | | Annotation Error | 1.84% | |
> > | **Advanced Practical Skills** <br> (IMG, ISD) | Factual Hallucination | 47.50% | In open-ended, generative tasks, models frequently invent facts, statistics, or product details to construct a plausible but incorrect response. This is a critical barrier to real-world deployment. |
> > | | Knowledge Gap | 19.08% | |
> > | | Poorly Formulated Answer | 15.37% | The answer, while not factually incorrect, is presented poorly—lacking professional tone, clarity, or empathy required for customer interaction. |
> > | | Others | 2.31% | |
> > | | Annotation Error | 1.47% | |
> >
> >
> > This analysis reveals that the primary failure modes are not simple knowledge gaps but more nuanced issues like Flawed Reasoning and Knowledge Confusion.
> >
> > **Action**: We will add a new "Error Analysis" subsection to Appendix, featuring this table and accompanied by detailed, fine-grained case studies for each major error type, as you suggested.
> >
> > **3. Regarding Weakness: "The author claims that only 10% of the data will be public..."**
> >
> > We recognize that limiting data access is a significant barrier to reproducibility and thank you for raising this critical point. In direct response to your feedback, we have taken two major steps:
> >
> > a) Increased Data Release: We have successfully arranged to increase the publicly available portion of our dataset **from 10% to 20%**. This significantly lowers the barrier for researchers to verify our results and benchmark their own models. To put this number into perspective, this 20% subset comprises nearly 9000 high-quality examples. Crucially, in absolute terms, our public dataset is larger than the entirety of many existing and widely used domain-specific benchmarks.
> >
> > b) Ensured Representativeness: This 20% subset is not a random sample but a carefully constructed, representative subset created using stratified sampling, preserving the original distribution of tasks and topics. To empirically prove its validity as a proxy for the full benchmark, we compared model performance on both sets.
> >
> >
> > | Model | Full Dataset (100%) Performance | Public Subset (20%) Performance | Δ (Difference) |
> > | :--- | :---: | :---: | :---: |
> > | Qwen3-32B | 81.92 | 81.00 | 0.92 |
> > | deepseek-r1 | 82.52 | 81.54 | 0.98 |
> > | gemini-2.5 | 83.89 | 84.09 | -0.20 |
> > | gpt-4o | 79.09 | 79.02 | 0.07 |
> >
> > As the table shows, the mean scores are highly consistent. The decision to not release 100% of the data stems from strict commercial data and compliance constraints, following a thorough internal legal review.
> >
> > **Action**: We will update Section 7 and the Data Availability Statement to reflect this increased 20% data release and include the validation table to assure the community of the subset's representativeness.
> >
> > We are confident that these substantial revisions and new analyses fully address the weaknesses identified. Thank you once again for your guidance, which has helped us to significantly improve the depth and contribution of our paper.

---

### Meta-Review · Area_Chair_JEH1 · 2026-01-07

**Summary:**

The reviewers identified critical flaws regarding the benchmark's longevity, reproducibility, and depth of analysis, which significantly limit its contribution to the ICLR community. During the rebuttal phase, the authors made great progress in supplementing the new experimental results and clarifying the data disclosure information. However, after the cautious evaluation, based on the reviews and the paper's content, the suggested decision is rejection. I encourage the authors to fully open the dataset and consider reviewers' comments in their next submission.

**Reviewer Concerns:**

1) Data Availability and Reproducibility:

A primary concern is the authors' statement that only 10% of the dataset will be made public due to commercial constraints. Reviewers argue this creates a significant barrier to reproducibility, making the paper's main claims unverifiable and severely reducing the benchmark's utility for the broader research community. Although the authors claim in the rebuttal that they have successfully arranged to increase the publicly available portion of our dataset from 10% to 20%.  As a scientific paper, this contribution cannot support its scientific value and should be reconsidered.

2) Benchmark Saturation and Difficulty:

Reviewers noted that current state-of-the-art models already achieve very high performance on the benchmark, with average scores reaching ~84% across most tasks. This high baseline performance suggests the benchmark lacks sufficient "headroom" for future improvement, raising doubts about its long-term utility in differentiating more capable future models.

3) Depth of Analysis:

The evaluation is criticized for being superficial. Multiple reviewers pointed out the absence of detailed error analysis, failure mode breakdowns, or fine-grained case studies.

**Reviewer Scores:**

Reviewer 9Dcj: Maintain the overall score as 4.

Reviewer Hewh: Maintain the overall score as 4.

Reviewer GsM6: Maintain the overall score as 4.

Reviewer 5LZM: Increase the overall score to 4.

---

### Decision · Program_Chairs · 2026-01-26

Reject